# A novel angiotensin I-converting enzyme inhibitory peptide derived from the trypsin hydrolysates of salmon bone proteins

**Thanakrit Kaewsahnguan**[1], **Sajee Noitang**[2], **Papassara Sangtanoo**[2], **Piroonporn Srimongkol**[2], **Tanatorn Saisavoey**[2], **Onrapak Reamtong**[3], **Kiattawee Choowongkomon**[4], **Aphichart Karnchanatat**[2]*

1 Program in Biotechnology, Faculty of Science, Chulalongkorn University, Pathumwan, Bangkok, Thailand, 2 Research Unit in Bioconversion/Bioseparation for Value-Added Chemical Production, Institute of Biotechnology and Genetic Engineering, Chulalongkorn University, Pathumwan, Bangkok, Thailand, 3 Department of Molecular Tropical Medicine and Genetics, Faculty of Tropical Medicine, Mahidol University, Ratchathewi, Bangkok, Thailand, 4 Department of Biochemistry, Faculty of Science, Kasetsart University, Chatuchak, Bangkok, Thailand

* Aphichart.K@chula.ac.th

**Data Availability Statement:** All relevant data are within the paper and its Supporting information files.

## Abstract

When fish are processed, fish bone becomes a key component of the waste, but to date very few researchers have sought to use fish bone to prepare protein hydrolysates as a means of adding value to the final product. This study, therefore, examines the potential of salmon bone, through an analysis of the benefits of its constituent components, namely fat, moisture, protein, and ash. In particular, the study seeks to optimize the process of enzymatic hydrolysis of salmon bone with trypsin in order to produce angiotensin-I converting enzyme (ACE) inhibitory peptides making use of response surface methodology in combination with central composite design (CCD). Optimum hydrolysis conditions concerning DH (degree of hydrolysis) and ACE-inhibitory activity were initially determined using the response surface model. Having thus determined which of the salmon bone protein hydrolysates (SBPH) offered the greatest level of ACE-inhibitory activity, these SBPH were duly selected to undergo ultrafiltration for further fractionation. It was found that the greatest ACE-inhibitory activity was achieved by the SBPH fraction which had a molecular weight lower than 0.65 kDa. This fraction underwent further purification using RP-HPLC, revealing that the $F_7$ fraction offered the best ACE-inhibitory activity. For ACE inhibition, the ideal peptide in the context of the $F_7$ fraction comprised eight amino acids: Phe-Cys-Leu-Tyr-Glu-Leu-Ala-Arg (FCLYELAR), while analysis of the Lineweaver-Burk plot revealed that the FCLYELAR peptide can serve as an uncompetitive ACE inhibitor. An examination of the molecular docking process showed that the FCLYELAR peptide was primarily able to provide ACE-inhibitory qualities as a consequence of the hydrogen bond interactions taking place between ACE and the peptide. Furthermore, upon isolation form the SBPH, the ACE-inhibitory peptide demonstrated ACE-inhibitory capabilities *in vitro*, underlining its potential for applications in the food and pharmaceutical sectors.

**Funding:** This study received support from the following sources: Thailand Science Research and Innovation (TSRI) Fund (CU_FRB640001_01_61_1), and the Center of Excellence on Medical Biotechnology (CEMB), S&T Postgraduate Education and Research Development Office (PERDO), Office of Higher Education Commission (OHEC), Thailand (SN-63-009-01), awarded to AK. The authors would like to make clear their gratitude for the assistance provided by the aforementioned bodies.

## Introduction

One of the major causes of death around the world today is cardiovascular disease (CVD). This is a type of disease that primarily damages the blood vessels and the heart, and its initial symptoms and risk factors include hypertension and high blood pressure. At the turn of the century, as many as 972 million people globally were though to suffer hypertension, with that figure anticipated to climb above 1.56 billion by 2025 [1]. Hypertension is designated as a chronic condition, and has been linked to problems in the metabolism which include resistance to insulin, abdominal obesity, diminished glucose tolerance, low levels of high density lipoprotein-cholesterol (HDL), along with hyperlipidemia and hyperglycemia [2, 3]. Blood pressure and balance of fluids will normally be governed by the renin-angiotensin system, which is a hormone system that is highly influential in CVD pathophysiology, especially in the case of congestive heart failure and also hypertension. It is the plasma renin which converts angiotensinogen which comes from the liver into angiotensin I; proteolytic cleavage then takes place when angiotensin I is exposed to angiotensin I-converting enzyme (ACE), resulting in the formation of angiotensin II in the lungs. Angiotensin II acts as a vasoconstrictor and therefore causes blood pressure to rise, while ACE has the effect of breaking down bradykinin, which would normally serve to maintain blood pressure at a lower lever through the promotion of vasodilatation. These properties explain why ACE inhibitors are likely to have a positive effect on patients with hypertension, and may present further benefits in reducing CVDs [4, 5].

The original synthesis of ACE inhibitors involved the compounds which are obtained from the venom of pit vipers, but today, synthetic ACE inhibitors are widely used as a means of controlling hypertension in human patients. Common types include lisinopril, enalapril, captopril, and ramipril. While these synthetic drugs are undoubtedly an effective form of treatment, there remains a disadvantage to their use since they can create adverse consequences when they are used for long periods of time. Such effects include rashes, allergic reactions, coughing, and problems affecting the sense of taste [6–8]. As a consequence, attempts to find a natural source of ACE inhibitors which can be effective while avoiding the adverse side effects have drawn increasing interest from researchers. It is hoped that synthetic drugs might be replaced, especially when taking into consideration the fact that increased consumption levels for fruit and vegetables is associated with a decline in chronic health conditions including cancer, CVD, inflammation, and problems connected to the ageing process [9]. Proteins which may serve to lower the risk of hypertension are one key area of interest with this goal in mind. The biologically active peptides represent a variety of peptides that differ in their amino acid compositions and arrangements. Many functions are reported, such as antioxidative, immune stimulating, hormone-modulating, antiviral, and antithrombotic [10–12].

Salmon bones, unwanted waste from fishing and fish processing, contain high protein content in the form of gelatin, collagen, and amino acids, which can serve as an important source of bioactive peptides, which in turn can be utilized to exploit their potential health benefits. In particular, collagen and gelatin have relatively high protein content and are already well-established as ingredient in the food and pharmaceutical sectors, as well as in the manufacture of cosmetics [13, 14]. This would indicate the potential for utilizing salmon bone, but to date its functional properties have limited its application. While the muscles and certain other waste forms of salmon have provided numerous bioactive peptides, there has been very little work conducted to date which examines the potential for extracting bioactive peptides from salmon bones.

Our previous research has discussed the anti-inflammatory and free radical scavenging activity of salmon bone protein hydrolysates in a lipopolysaccharide (LPS)-stimulated

macrophage model, with our results indicating that the release of nitric oxide (NO) by the peptide fraction was limited as a result of decreased expression of proinflammatory cytokines gene expression. It was also noted that the salmon bone amino acids comprised the positively charged glycine, alanine, arginine, and proline [15]. In this research, certain variables were chosen for optimization in order to determine the degree of hydrolysis when producing the protein isolate from salmon bones. RSM (response surface methodology) was applied in order to determine the optimal hydrolysis conditions in terms of temperature, pH value, and E/S ratio. To select one with high ACE inhibitory activity, protein hydrolysate was purified, identified, and synthesis of newly isolated ACE inhibitory peptides was performed; their $IC_{50}$ values were determined, and peptide inhibition patterns were examined using Lineweaver-Burk plots. Finally, to measure the affinities of various chemical compounds for ACE, molecular docking served to demonstrate the ACE mechanism of inhibition.

## Materials and methods

### Biomaterials and chemicals

Atlantic salmon bones were used for the study, which were obtained from Oishi Group Public Co., Ltd. (Bangkok, Thailand). This product was stored at -20˚C until needed, after which the meat was removed from the bones by a boiling process. The clean bones were then dried in a hot air oven at a temperature of 60˚C for 24 hours, after which the bones were ground into powder using a blender, which was then sieved through a 150 μm sieve. The powder was then filled into polypropylene plastic and stored at room temperature in a desiccator until required.

The chemicals used in the study included ACE (E.C. 3.4.15.1) derived from rabbit lungs, acetonitrile, bovine serum albumin (BSA), captopril, dithiothreitol (DTT), hippuric acid, hippuryl-L-histidyl-L-leucine (HHL), o-phthaldehyde (OPA), sodium dodecyl sulfate (SDS), sodium tetraborate decahydrate, 2, 4, 6-trinitrobenzenesulfonic acid (TNBS), and Trypsin from porcine pancreas (1500 U/mg), all of which were obtained from Sigma Chemical Co. (St. Louis, MO, USA). In addition, dithiothreitol (DTT) and L-serine were purchased from Merck (Germany). All chemicals used were of analytical grade.

### Proximate analysis

The approach recommended by the Association of Official Analytical Chemists (AOAC) was employed to investigate the proximate composition for the samples of salmon bone powder. Initially, a vacuum oven (AOAC 925.09, and 926.08) [16] was used to measure the sample moisture content before the lyophilization process. All samples were safely sealed in containers and underwent re-suspension after they had been thawed to ensure that no losses would be incurred. AOAC 923.03 was used as the guide to measuring the total ash content [17], while AOAC 2003.06 was used for measuring total fat [18]. Automated Kjeldahl apparatus was used to measure the nitrogen content in line with AOAC 968.06, and 992.15, allowing the overall protein content to be subsequently obtained through multiplication of the nitrogen content by 6.25 [19]. Each of these analyses was carried out in triplicate.

### Enzymatic hydrolysis of the salmon bone proteins

Three key hydrolysis variables were tested in the preparation of SBPH: temperature, time, and E/S (enzyme to substrate) ratio. Under the experimental design, 2.5 g of salmon bone powder was added to 50 mL of phosphate-buffered saline (PBS; 20 mM phosphate buffer containing 0.15 M NaCl pH 7.2), after which the pH was gradually adjusted to an optimal value of 8.0 by adding 0.5 M NaOH. Subsequently, the hydrolysis reaction was carried out with trypsin in a

shaking incubator at 150 rpm. To complete the reaction, the mixture was heated to a temperature of 90°C for 20 minutes to inactivate the enzyme. The final step was centrifugation at 19,230 × g for 30 minutes at a temperature of 4°C before collecting the supernatant, which was then stored at -20°C until use.

## Experimental design for the optimization of SBPH enzymatic hydrolysis conditions via RSM

The conditions for the enzymatic hydrolysis of the salmon bones were optimized via RSM, with the effects on the response variables of the independent variables being estimated by RSM in combination with CCD (central composite design). The experimental design made use of temperature (A), time (B), and E/S ratio (C) as the independent variables while the selected response variables (Y) comprised the degree of hydrolysis ($Y_1$) and the ACE-inhibitory activity ($Y_2$). For the three independent variables, the CCD involved five coded levels: −1.68, −1, 0, +1, and +1.68. Twenty experiments were conducted, all in triplicate, comprising 8 factorial points, 6 axial points, and 6 replicates of the central point. Design and statistical analysis were carried out using Design Expert Software Version 12 (Stat-Ease, Inc., USA). In order to evaluate the influence of the independent variables upon the response variables, the second-order polynomial regression model was selected for analysis.

## Degree of hydrolysis (DH)

The SBPH was subjected to evaluation to determine the degree of hydrolysis using the approach proposed by Nielsen *et al*. [20]. Briefly, 3 mL of the prepared OPA regent was added to a test tube containing 400 mL of SBPH. Mixing was performed for 5 seconds, after which the mixture was allowed to stand at room temperature for 2 minutes before detection at 340 nm was performed using a spectrophotometer. The standard used was serine, which was prepared by dissolving 50 mg of serine in 500 mL of deionized water. The assays were all carried out in triplicate. DH was then determined using the formula given below in Eq (1):

$$\%\mathrm{DH} = (h/h_{tot}) \times 100 \tag{1}$$

in which $h_{tot}$ indicates the overall number of peptide bonds found in the salmon protein substrate (7.501 mequiv/g), and $h$ represents the total number of hydrolyzed bonds (h = (serine-$NH_2$ - β) / α meqv/g protein). The values for β and α rely upon the composition of the amino acids which serve as raw materials. Eq (2) was then employed for the calculation of serine-$NH_2$:

$$\mathrm{Serine\text{-}NH_2} = \left(\mathrm{OD_{sample}} - \mathrm{OD_{blank}}\right)/(\mathrm{OD_{standard}} - \mathrm{OD_{blank}}) \times 0.9516 \ \mathrm{meqv/L} \times 0.05 \times 100/\mathrm{X} \times \mathrm{P} \tag{2}$$

in which $\mathrm{OD_{sample}}$ indicates sample absorbance at 340 nm, $\mathrm{OD_{blank}}$ indicates water absorbance at 340 nm, $\mathrm{OD_{standard}}$ indicates serine absorbance at 340 nm, X represents the sample quantity in g, P represents the protein percentage in the salmon bone (41.05%), and the total sample volume used was 0.05 L.

## Protein content determination

Protein concentration in supernatants was determined by Bradford's method [21]. Bovine serum albumin (BSA) was used as a protein standard.

## ACE-inhibitory activity assay

The ACE -inhibition test was performed following the approach of Ibrahim *et al.* [22]. At the beginning of the inhibition assay, a 10 µL aliquot of the peptide sample was added to 31 µL of 50 mM potassium phosphate buffer with a pH of 8.3 containing 0.3 M NaCl in each well of the 96-well microplate, while PBS was used for the control reaction. In the next step, a 5 µL aliquot of 200 mU/mL ACE solution was added to each of the wells and the reaction was then initiated by adding 13 µL of the substrate (5 mM HHL solution) and shaking the mixture before incubation at a temperature of 37˚C for one hour. Subsequently, 50 µL of 10 mM sodium sulfite at a pH of 9.11, 100 µL of 200 mM potassium phosphate buffer at a pH of 6.77, and 50 µL of 0.68 mM TNBS were added to each of the wells, and the mixtures were then incubated at 37 ˚C for 20 minutes. The ACE -inhibitory activity was determined by measuring the absorbance at 420 nm. For each of the samples, the assay was performed in triplicate, with captopril serving as a positive control. The following Eq (3) was used to calculate the percentage of ACE -inhibitory activity:

$$\text{ACE-inhibitory activity } (\%) = [(C - Bi) - (S - Bs)/(C - Bi)] \times 100 \tag{3}$$

in which the absorbance of the control is given by C (100% activity), the sample (inhibitor peptide) is given by S, the blank inhibitor (HHL alone) is given by Bi, and the blank sample (peptide alone) is represented by Bs. The $IC_{50}$ value could then be determined as the ACE inhibitor concentration which would result in a 50% decrease in ACE activity during the assay. Its calculation is based on non-linear regression and can be performed using GraphPad Prism Version 6.01 (GraphPad Software Inc., La Jolla, CA, USA).

## Fractionation and purification of ACE-inhibitory peptides

**Ultrafiltration.** After identification of SBPH with the greatest ACE -inhibitory activity, further fractionation was performed with an ultrafiltration unit (MinimateTM TFF Capsule) (Pall Corporation, USA) using four MWCO (molecular weight cut-off) membranes of 10, 5, 3 and 0.65 kDa. In the first step, the hydrolysates were passed through the 10 kDa membrane to give the retentate (10 kDa fraction) and permeate. In the next step, the permeate was subjected to further filtration using the 5 kDa membrane, again yielding a retentate (5–10 kDa fraction) and permeate. Further retentates were obtained using the 3 kDa and 0.65 kDa membranes, resulting in a set of five fractions classified by molecular weight. The fractions were then assayed for their protein concentrations and ACE -inhibitory activity, after which the fractions that offered the greatest potential for ACE -inhibition were selected for the next step, in which their concentration was increased in Refrigerated Centrivap -vacuum concentrators before being stored at -20˚C until the subsequent purification process.

**Reversed Phase High Performance Liquid Chromatography (RP-HPLC).** The fraction with the greatest ACE -inhibitory activity was subjected to further purification by RP-HPLC (Spectra System, Thermo Fisher Scientific) using a Luna 5U C18 100 A˚ column (4.6 × 250 mm, Luna 5 µm; Phenomenex, Torrance, CA). A nylon filter with a polypropylene housing and a pore size of 0.45 µm (Whatman, GE, Buckinghamshire, UK) was used to filter 1 mL of this fraction into a vial before loading 100 µL into the column of the HPLC system. Gradient elution was then performed using 100: 0% (v/v) of mobile phase A: B set at 88: 12 at the 15 minutes mark, followed by 75: 25 at 30 minutes, then 65: 35 at 35 minutes, followed by 60: 40 at 40 minutes and finally 55: 45 at 50 minutes, with a flow rate of 0.7 mL/min. Mobile phase A consisted of 0.1% (v/v) trifluoroacetic acid (TFA) in distilled water, while mobile phase B consisted of 70% (v/v) acetonitrile (ACN) in 0.05% (v/v) TFA. The absorbance of the effluent fraction was evaluated at 280 nm, while the different fractions associated with each of the peaks

were collected for further concentration before their ACE-inhibitory activities could be evaluated.

## Amino acid sequence identification for the purified peptide and *de novo* peptide sequencing

The RP-HPLC fraction which demonstrated the greatest ACE-inhibitory activity underwent further analysis to determine the molecular mass and amino acid sequence using an Ultimate® 3000 Nano-Liquid Chromatography Systems (Thermo Scientific™ Dionex™) in combination with a micrOTOF-Q II™ ESI-Qq-TOF mass spectrometer (Bruker Daltonics, Germany). Hystar™ software (Bruker Daltonics) was used to manage the data obtained from the mass spectrometer. The MS and MS/MS spectra encompassed a mass range from m/z 50–2000. The process of *de novo* sequencing was then carried out with the LC-MS/MS data. Mascot database software (Matrix Science, London, UK) was then used to further examine the MS data files in the context of the NCBI database. The amino acid sequence which resulted was then explored further using the NCBI database using the BLASTp program as a means of protein identification (ttps://blast.ncbi.nlm.nih.gov/Blast.cgi). Through the use of the BLASTp program it is possible to match the observed amino acid sequences to those of the databases and to determine whether those matches are statistically significant.

## Peptide synthesis

Having selected a suitable ACE-inhibitory peptide from the SBPH, the peptide then underwent chemical synthesis via Fmoc solid phase using an Applied Biosystems Model 433A Synergy peptide synthesizer (Applied Biosystems, Foster City, CA, USA). Analytical mass spectrometry was then performed to determine the peptide purity using a quadrupole ion trap Thermo Finnigan™ LXQ™ LC-ESI-MS (San Jose, CA, USA) connected to a Surveyor HPLC (Thermo Fisher Scientific, San Jose, CA, USA). The purity of the synthetic peptide was reported to be at least 95% on the basis of the results from HPLC analysis. The peptide sequence was FCLYELAR, while the molecular weight was 1014.20 Da, with 97.3% HPLC purity. The $IC_{50}$ value for the synthetic peptide was then measured using the approach described earlier.

## Bioinformatics search of the SBPH-derived ACE-inhibitory peptides conducted *in silico*

The BIOPEP database (http://www.uwm.edu.pl/biochemia/index.php/pl/biopep) was used in order to make the predictions concerning the potential activity of the sequences identified from SBPH (FCLYELAR) in terms of ACE inhibition, while ToxinPred served to make the predictions for the *in silico* toxicity levels of these SBPH ACE-inhibitory peptides (https://webs.iiitd.edu.in/raghava/toxinpred/index.html). A support vector machine (SVM) prediction technique was applied with a threshold value set to 0.0 as a means of classifying peptides on the basis of toxicity or non-toxicity. Peptide solubility was then assessed using the Innovagen server (www.innovagen.com/proteomics-tools), while the sensory evaluation of the peptides was completed with the previously mentioned BIOPEP tool.

## ACE inhibition kinetics

ACE-inhibition kinetics were evaluated using varied HHL concentrations (0.5, 1, 3, 5, and 7 mM) to which varied FCLYELAR peptide concentrations were introduced (0.05, 0.1 and 0.2 mM). The reaction assay required that the peptide concentrations which had been dissolved in PBS undergo incubation with the ACE solution which contained the different HHL

concentrations which had been dissolved in the reaction buffer. Aliquots of sodium sulfite and TNBS were subsequently added prior to 80minutes of incubation at a temperature of 37˚C, allowing the level of ACE-inhibitory activity to be evaluated. Lineweaver-Burk plots were used to assess the inhibition kinetics in the presence or absence of the peptide, whereby the concentration for HHL was represented along the x-axis, while the y-axis showed the absorbance at 420 nm (HL+TNBS complex). Secondary plots were then used to determine the inhibitor constant (Ki). Accordingly, when 1/V max is plotted against the peptide concentration, the result is linear, while the intercept on the axis representing the inhibitor concentration was considered to be Ki.

## Molecular docking process

Simulations of the molecular docking process involving the peptides and ACE were carried out with the GOLD program version 5.7.1 software employing a modified version of the approach recommended by Kheeree *et al.* [23]. For this part of the study, the 3D crystal structure of the Angiotensin Converting Enzyme (PDB code: 1O8A) was obtained from the RCSB Protein Data Bank (PDB) (https://www.rcsb.org/structure/1o8a) to serve as the receptor in the docking experiments, while the structure of the FCLYELAR peptide was produced using Discovery Studio Visualizer v19.1.0 software, to create a MOL2 file. To prepare the ACE structure, the water molecules were first of all removed, and hydrogen atoms were added to the ACE model using the GOLD setup process. The docking site is positioned on the ligand located at the active site of the ACE structure inside a 6 Å radius. A Genetic Algorithm (GA) search process was then performed at most accurate level (slow mode) for a total of 30 runs, and the molecular docking was then evaluated using the Chem Score fitness function. To rank the docking of the purified peptides at the ACE binding site, the PLP fitness scores were employed, while Discovery Studio Visualizer v19.1.0 software identified the H-bonds and the various hydrophilic, hydrophobic, electrostatic, and van der Waals forces which arise as the ACE molecules and peptide residues interact.

## Statistical analysis

Each experiment in the study was performed in triplicate, with results presented in the form of mean ± standard deviation (SD). Statistical analysis used IBM SPSS statistics version 22, while hypotheses were tested using one-way ANOVA (F-test). To assess the statistical significance of differences between means, Post-Hoc multiple comparisons were used via the Duncan test, and the level of significance was set at $P < 0.05$.

# Results and discussion

## Proximate analysis

Table 1 shows the proximate salmon bone powder composition. Upon analysis, the powder was confirmed to contain only minimal quantities of moisture, indicating the effectiveness of the drying procedure. Reports have indicated that water molecules which were present had not permeated into the tissue of the bone, but were found only on the surface, so the drying process in an oven would remove almost all of the water. Due to its low moisture content, the powder would then be relatively stable, even at room temperature. Furthermore, no aggregation would take place when the powder was held in storage. This low moisture content also helped to protect the bone powder from microbial damage, since moisture levels of 2% or lower cannot support the growth of microbes [24, 25]. As a

**Table 1. Principal contents of salmon bone powder.**

| Parameter | Proximate in dry weight (%w/w) |
|---|---|
| Moisture | 2.18 + 0.06 |
| Protein | 39.96 + 0.28 |
| Fat | 2.04 + 0.20 |
| Ash | 51.65 + 0.12 |

Mean ± SE calculated from data recorded in triplicate.

consequence of its stability, fish bone powder could be readily used as a key ingredient for various purposes.

Salmon bone powder comprised around 40% protein, which generally compares favorably with the protein content levels for other fish bone powders which lie in the range of 26% to 41% for mackerel, saithe, trout, cod, blue whiting, and herring [24]. These results might be influenced by the kind of preparation performed on the fish, since some studies took the flesh from the bone using boiling water, while other studies applied a hot alkaline solution to perform the same task. It would appear that the advantage of using the alkaline solution method is that greater quantities of flesh and protein can be extracted from the bones, although alkaline solution is far from perfect in this regard since it cannot remove all of the protein [26]. Some of the protein will remain in the bone powder. The proteins found in the bone are usually classified into stroma protein. Meanwhile, it has been reported that the bone powder from mackerel, blue whiting, and herring contains various amino acids along with glycine, proline, and hydroxyproline [27].

It is also understood that lipids are contained within fish bone, especially in the main bone framework which has joints with many of the other minor bones. Due to their complexity, it is difficult to remove these lipids simply by immersing them in the alkaline solution. The lipid content has been reported in the range of 1% to 27%, while in the case of mackerel it can be as high as 47%, whereas in saithe and cod it is less than 2%. It is also likely that fat residue will be found in the fish bone powder, especially given that crude fat amounts to around 6% in the original bone tissue [26]. For each different fish species, the lipids found in bone are correlated with the levels of body fat, noting that particularly large or old fish tend to have higher levels of fat content. It has also been shown that the fish bone powder obtained from a range of species contains around 80% unsaturated fatty acids [28]. In usual circumstances, unsaturated fatty acids will readily undergo oxidative degradation, but this is less likely in the case of salmon bone powder due to the low levels of fat content.

Bone powder is 51.65% ash, although in some fish species the ash proportion starts at 40%. In order to ensure a high degree of bone component purity, it is therefore necessary to carry out bone preparation. Hardness is a key quality, which depends on the degree of bone mineralization, which in turn is based on the ratio of ash to protein; in the case of salmon bone this ratio stands at 1.29, although this can be influenced by the preparation process [26]. The major bone component is calcium, which occurs in various different forms, including calcium citrate, calcium phosphate, and calcium acetate. Kadam and Prabhasankar [29] note the significance of fishbone as an important calcium source in which the calcium and phosphorus are present in balance. Furthermore, fish are also a key source of minerals such as iron, copper, and zinc, which are needed for the hormones and enzymes to function effectively. The bones do, however, carry a greater proportion of ash than any other part of the body, and this value may also be indicative of the bone powder purity [30, 31].

## Optimization of salmon bone hydrolysis

**Preliminary assessment.** RSM was used in the initial testing process to find the optimal conditions for the hydrolysis of SBPH whereby the various factors were altered in isolation in order to determine their influence on the outcome. These factors included hydrolysis time, temperature, and E/S ratio, and the aim was to assess the center point values of these varied hydrolysis parameters (S1 Table). For instance, the DH (degree of hydrolysis) and $IC_{50}$ (ACE-inhibitory activity) values for the SBPH were tested at a range of temperatures from 30°C to 60°C while holding the hydrolysis time at 180 minutes, the pH value at 8.0, and the E/S ratio at 0.4% (w/w). The outcomes are presented in Fig 1: in Fig 1(a), a change in temperature from 30°C to 40°C saw the DH value rise while the $IC_{50}$ value decreased notably. Once the temperature rose above 40°C, there was a gradual rise in the DH value, while the $IC_{50}$ value also showed a slight rise. It is understood that at higher temperatures, the proteases are deactivated, causing enzymatic hydrolysis to be incomplete, and thus leading to reduced values for DH and ACE-inhibitory activity [32]. The optimal temperature based on these findings was 40°C since this gave the highest percentage for DH along with a low value for $IC_{50}$, and according this temperature was selected as the center point (0) for subsequent testing, as indicated in Table 2.

The next series of tests examined the influence of a range of hydrolysis times between 120 and 360 minutes. The temperature was set to 40°C, with an E/S ratio of 0.4% (w/w), and a pH value of 8.0, as shown in Fig 1(b). The effects upon DH and $IC_{50}$ were then evaluated. It was found that DH increased as hydrolysis time increased, while the opposite was true for the $IC_{50}$ value which dropped significantly as the hydrolysis time was extended, thus indicating stronger ACE-inhibitory activity. Accordingly, 360 minutes was selected as the duration for further tests, in concurrence with the work of Gao *et al.* [33], who reached similar conclusions when studying the hydrolysis of cottonseed protein using papain, advocating an optimal hydrolysis time of 360 minutes, while noting that a longer time might be counter-productive since it may adversely affect the ACE-inhibitory ability of the peptides. This was a view shared by Mullally *et al.* [34], who reported no further improvement in ACE-inhibitory activity as a result of extending the hydrolysis period. A time of 360 minutes was therefore selected as the center point (0) for subsequent testing, as indicated in Table 2.

In Fig 1(c), the E/S ratio was altered from 0.1% to 0.8% (w/w) while the hydrolysis time was fixed at 180 minutes, the temperature at 40°C, and the pH value at 8.0. The effects on DH and $IC_{50}$ were then measured, revealing that as the E/S ration increased, the DH value also steadily increased. However, the ACE-inhibitory activity, indicated by $IC_{50}$, was maximized at an E/S value of 0.4% (w/w) while further increases in the E/S ratio caused performance to deteriorate. These findings matched those of Guo *et al.* [35] whose studies showed that an increasing E/S ratio would lead to a rise in DH, but would not improve ACE-inhibitory activity. It can be explained that the rise in DH would result from increased hydrolysis of the protein upon the addition of further proteases [34]. Accordingly, for the further experiments in this study, and E/S ratio of 0.4% (w/w) was selected as the center point (0) as shown in Table 2.

**RSM fitting.** Similar to the preliminary investigation, RSM based on CCD was employed to determine the optimal hydrolysis conditions in order to produce the ACE-inhibitory peptides. The model was established by assessing the impact of temperature (A), time (B), and E/S ratio (C), which served as the independent variables from the salmon bones and was employed to establish the model by studying the effect of three independent variables including temperature (A), time (B), and E/S ratio (C), upon the degree of hydrolysis and ACE-inhibitory activity which served as the response variables indicated respectively by Y1 and Y2 (S2 and S3 Tables). From the earlier experiments, the center point values for the three independent variables had already been set at 40°C, 360 minutes, and 0.4% (w/w). Table 3 shows the outcomes for the

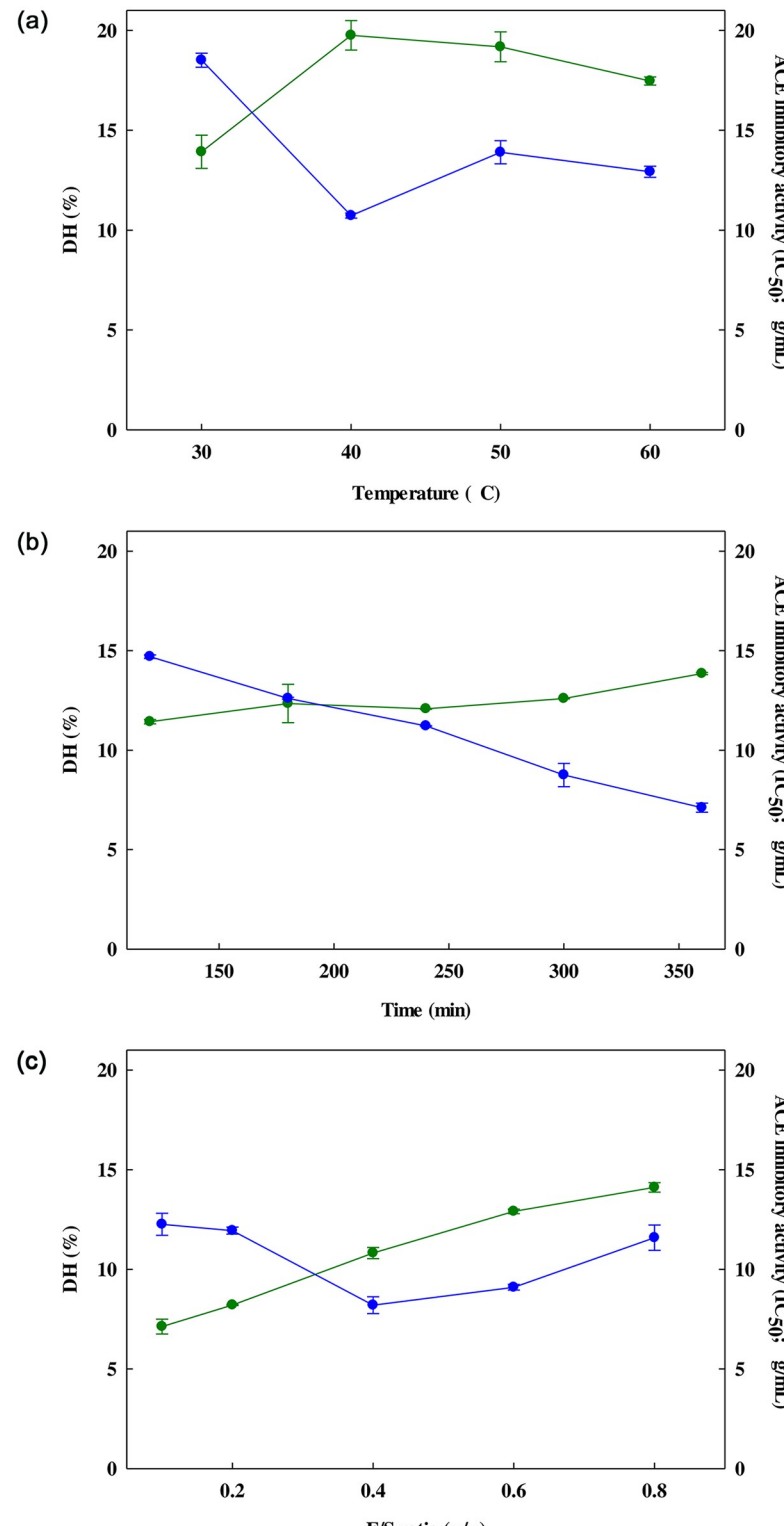

**Fig 1. Influence of temperature (a), hydrolysis time (b), and E/S ratio (c) upon DH (●) and ACE-inhibitory activity (●) of the hydrolysate.** Each data value is presented in the form of mean ± S.D. and the tests were conducted in triplicate.

Table 2. Independent variables and code levels for the SBPH based on CCD.

| Independent variables | Code units | Coded Levels | | | | |
|---|---|---|---|---|---|---|
| | | -1.68 | -1 | 0 | +1 | +1.68 |
| Temperature (˚C) | A | 28.24 | 33 | 40 | 47 | 51.76 |
| Time (minutes) | B | 259.2 | 300 | 360 | 420 | 460.8 |
| E/S ratio (% w/w) | C | 0.064 | 0.2 | 0.4 | 0.6 | 0.736 |

response variables from 20 experimental runs based on CCD in varying conditions. For the SBPH, the DH values were in the range of 8.92% to 17.56% and can be seen in Table 3, with the highest value derived from the eighth experimental run. In a study of DH values for salmon heads, Gbogouri *et al.* [36] reported similar findings, with values ranging from 11.5% to 17.3%. Table 3 also shows the ACE-inhibitory activity, with results in the range of 6.87 to 12.55 µg/mL. Run 19 provided the lowest experimental value for ACE-inhibitory activity at 6.87 µg/mL. The ANOVA results used p-values lower than 0.05 along with the estimated regression coefficients from the second-order polynomial model describing degree of hydrolysis ($Y_1$) or ACE inhibition ($Y_2$), derived respectively from Eqs (4) and (5). Table 4 presents the DH regression coefficients which indicate that the most significant effects resulted from the linear terms of temperature (A), time (B), and E/S ratio (C) ($p < 0.0001$ or $p < 0.05$) while the interactions among those variables (AB, AC, BC) as well as the quadratic terms ($A^2$, $B^2$, $C^2$) were shown to have effects upon DH which were not significant ($p > 0.05$). These findings demonstrate that the greatest influence upon DH was exerted by the linear terms of temperature, E/S ratio, and time, while the regression coefficients associated with ACE inhibition

Table 3. Results of the CCD for the values of the two response variables, which encompass %DH ($Y_1$), and ACE inhibitory activity ($Y_2$) under various reaction conditions pertaining to the three independent variables (A, B, C).

| Run | Independent variables | | | Response (experimental) | |
|---|---|---|---|---|---|
| | A | B | C | $Y_1$ | $Y_2$ |
| 1 | 33 | 300 | 0.2 | 9.35 | 12.14 |
| 2 | 47 | 300 | 0.2 | 11.88 | 11.65 |
| 3 | 33 | 420 | 0.2 | 9.60 | 11.78 |
| 4 | 47 | 420 | 0.2 | 12.57 | 10.49 |
| 5 | 33 | 300 | 0.6 | 12.90 | 10.44 |
| 6 | 47 | 300 | 0.6 | 16.22 | 10.91 |
| 7 | 33 | 420 | 0.6 | 14.26 | 10.82 |
| 8 | 47 | 420 | 0.6 | 17.56 | 10.44 |
| 9 | 28.24 | 360 | 0.4 | 10.00 | 10.51 |
| 10 | 51.76 | 360 | 0.4 | 14.66 | 9.70 |
| 11 | 40 | 259.2 | 0.4 | 11.77 | 10.34 |
| 12 | 40 | 460.8 | 0.4 | 14.25 | 10.15 |
| 13 | 40 | 360 | 0.064 | 8.92 | 12.55 |
| 14 | 40 | 360 | 0.736 | 17.11 | 11.63 |
| 15 | 40 | 360 | 0.4 | 13.70 | 7.26 |
| 16 | 40 | 360 | 0.4 | 12.86 | 7.10 |
| 17 | 40 | 360 | 0.4 | 12.42 | 7.12 |
| 18 | 40 | 360 | 0.4 | 12.31 | 7.11 |
| 19 | 40 | 360 | 0.4 | 12.95 | 6.87 |
| 20 | 40 | 360 | 0.4 | 13.27 | 7.00 |

**Table 4. ANOVA and estimations of regression coefficients for DH ($Y_1$) and ACE-inhibition ($Y_2$) using the response surface quadratic model.**

| Source | df | Sum of Squares | | Mean Square | | F-value | | p-value | |
|---|---|---|---|---|---|---|---|---|---|
| | | $Y_1$ | $Y_2$ | $Y_1$ | $Y_2$ | $Y_1$ | $Y_2$ | $Y_1$ | $Y_2$ |
| Model | 9 | 106.65 | 72.55 | 11.85 | 8.06 | 55.87 | 330.92 | < 0.0001* | < 0.0001* |
| A | 1 | 29.19 | 0.69 | 29.19 | 0.69 | 137.62 | 28.28 | < 0.0001* | 0.0003* |
| B | 1 | 4.48 | 0.28 | 4.48 | 0.28 | 21.11 | 11.35 | 0.0010* | 0.0071* |
| C | 1 | 71.81 | 1.83 | 71.81 | 1.83 | 338.56 | 74.95 | < 0.0001* | < 0.0001* |
| AB | 1 | 0.02 | 0.34 | 0.02 | 0.34 | 0.10 | 14.06 | 0.7556 | 0.0038* |
| AC | 1 | 0.16 | 0.44 | 0.16 | 0.44 | 0.73 | 17.9 | 0.4124 | 0.0017* |
| BC | 1 | 0.39 | 0.25 | 0.39 | 0.25 | 1.83 | 10.46 | 0.2058 | 0.009* |
| $A^2$ | 1 | 0.33 | 16.75 | 0.33 | 16.75 | 1.56 | 687.85 | 0.2396 | < 0.0001* |
| $B^2$ | 1 | 0.11 | 18.33 | 0.11 | 18.33 | 0.51 | 752.72 | 0.4900 | < 0.0001* |
| $C^2$ | 1 | 0.12 | 45.70 | 0.12 | 45.70 | 0.56 | 1876.02 | 0.4727 | < 0.0001* |
| Residual | 10 | 2.12 | 0.24 | 0.21 | 0.02 | | | | |
| Lack of Fit | 5 | 0.76 | 0.16 | 0.15 | 0.03 | 0.56 | 1.88 | 0.7279 | 0.2526 |
| Pure Error | 5 | 1.36 | 0.08 | 0.27 | 0.02 | | | | |
| Cor Total | 19 | 108.78 | 72.79 | | | | | | |
| Std. Dev. | | 0.4605 | 0.1561 | | | | | | |
| Mean | | 12.929 | 9.8 | | | | | | |
| C.V. % | | 3.5621 | 1.59 | | | | | | |
| $R^2$ | | 0.9805 | 0.9967 | | | | | | |
| Adj $R^2$ | | 0.9630 | 0.9936 | | | | | | |
| Adeq Precision | | 26.589 | 51.182 | | | | | | |

* variables found to exert a significant effect upon the response ($p < 0.0001$ or $p < 0.05$).

revealed significant effects ($p < 0.0001$ or $p < 0.05$) on ACE-inhibitory activity for each of the linear (A, B, C), interaction (AB, AC, BC), and quadratic terms ($A^2$, $B^2$, $C^2$).

Model fitness was assessed through the application of the coefficient of determination ($R^2$) in combination with the lack of fit test and probability (p) values. Table 3 presents the ANOVA results for the response surface quadratic model describing the degree of hydrolysis and the ACE-inhibitory activity which revealed F-values of 55.87 and 330.92, while the p-value ($p < 0.0001$) was very low, from which it can be inferred that the terms in the model were highly significant. Meanwhile, for the model of DH and ACE inhibition, the respective p-values of the lack of fit test were 0.7279 and 0.2526, which in terms of the significance level ($p > 0.05$) suggested that the model offered sufficient accuracy in predicting the values of DH and ACE-inhibitory activity for the independent variables studied in all combinations for the range of values tested. Moreover, the design of the lack of fit test aims to determine whether the experimental data can be satisfactorily described by the model, or whether it would be necessary to apply a model of greater complexity. The model of DH and ACE inhibition produced very high $R^2$ values of 0.9805 and 0.9967, demonstrating that the response variability would be very closely explained by the model according to Eqs (4) and (5). $R^2$ values closer to 1.00 are indicative of stronger ability for the model to predict the response with greater accuracy [37]. Meanwhile, the respective Adjusted $R^2$ values for DH and the ACE inhibition were 0.9630 and 0.9936, indicating that for DH, just 3.7% of the variation was not explained by the model, while for ACE-inhibitory activity, 0.64% was not explained.

Adequate precision indicates the predicted response range in terms of its associated error, or the ratio of signal to noise, whereby a value in excess of 4 is considered preferable. In this research study, for DH and ACE inhibition, the respective adequate precision ratios were

26.589 and 51.182, which represent acceptable results, so the models are both suitable for navigation of the design space [38]. Furthermore, data dispersal can be assessed using the coefficient of variation (C.V.), which shows the standard deviation as a percentage of the mean, and which exhibited values of 3.56% and 1.59% for DH and ACE-inhibition, respectively. Since they are low, these values are considered acceptable, and indicate relatively good accuracy for the models. Had the coefficient of variation been higher, this would suggest a high degree of variation, and thus a failure to produce a suitable response model.

Eqs (4) and (5) provide the second-order polynomial regression model which can be used to explain the effects of the independent variables upon the response variables, for which DH and ACE inhibition were represented respectively by $Y_1$ and $Y_2$. The equation parameter was determined from the experimental data by using multiple regression analysis and RSM.

$$Y_1 = 1.0865 + 0.3725 \ A - 0.0202 \ B - 0.9325 \ C + 0.0001 \ AB + 0.0995 \ AC + 0.0184 \ BC \\ - 0.0031 \ A^2 + 2.42 \times 10 - 5 \ B^2 + 2.2666 \ C^2 \tag{4}$$

$$Y_2 = 90.7283 - 1.6846A - 0.2146B - 49.5222C - 0.0005AB + 0.1667AC + 0.0149BC \\ + 0.0220A^2 + 0.0003B^2 + 44.5881C^2 \tag{5}$$

in which Y represents the values predicted for DH and ACE inhibition ($IC_{50}$). The letters A, B, and C respectively indicate the temperature, hydrolysis duration and E/S ratio.

**Response surface and contour plots.** The response surface plots (3D) and contour plots (2D) were produced from a quadratic equation whereby the z-axis is used to plot the response values of DH and $IC_{50}$ against any other pair of independent variables on the x- and y-axes, with the other independent variables held to a fixed value at the center point. These can be seen in Figs 2 and 3. The plots can help to provide a clear representation of the interactions which take place among independent variables and the effect upon the response variables. They can also be helpful in determining the optimal hydrolysate condition offering the best ACE inhibition and degree of hydrolysis. In Fig 2(a), the linear influence on DH of hydrolysis time and temperature is shown when the E/S ratio is held to a fixed value at 0.4%. The findings indicated that as the two independent variables increased, DH also increased. Fig 2(b) shows the effects of temperature and E/S ratio upon DH when the hydrolysis time is fixed at 360 minutes. In this case, an almost linear increase in DH resulted from the rising temperature and E/S ratio, in concurrence with the findings of Espejo-Carpio *et al*. [39] who found that greater E/S ratios resulted in hydrolysates which offered a higher degree of hydrolysis for all temperatures. Greater temperatures and enzyme concentrations resulted in the exposure of the peptide bonds, which readily underwent cleavage during the process of enzymatic hydrolysis, thus raising the degree of hydrolysis [40]. Finally, Fig 2(c) shows the effects of hydrolysis time and E/S ration on DH when the temperature was maintained at 40°C, revealing that as the independent variables increased, the degree of hydrolysis also increased. This was confirmed by Nilsang *et al*. [41] whose work examined the flavoenzyme hydrolysis of fish soluble concentrate with 5% enzyme concentration, reporting that at the greatest hydrolysis time of 360 minutes, the maximum DH value of 62% was obtained.

Fig 3(a)–3(c) presents the response surface plots for ACE inhibition ($IC_{50}$). The shape is broadly concave; from which it can be inferred that the conditions were optimal. Where the response surface plot presents circular contours, this implies that there is negligible interaction among the related variables. However, if the contour plots take an elliptical form, this shows that the interactions among related variables are relatively significant [42]. Fig 3 shows elliptical plots, suggesting significant interactions, which match the ANOVA results concerning the quadratic model. In Fig 3(a) it can be seen that as the temperature increases from 28.24°C to

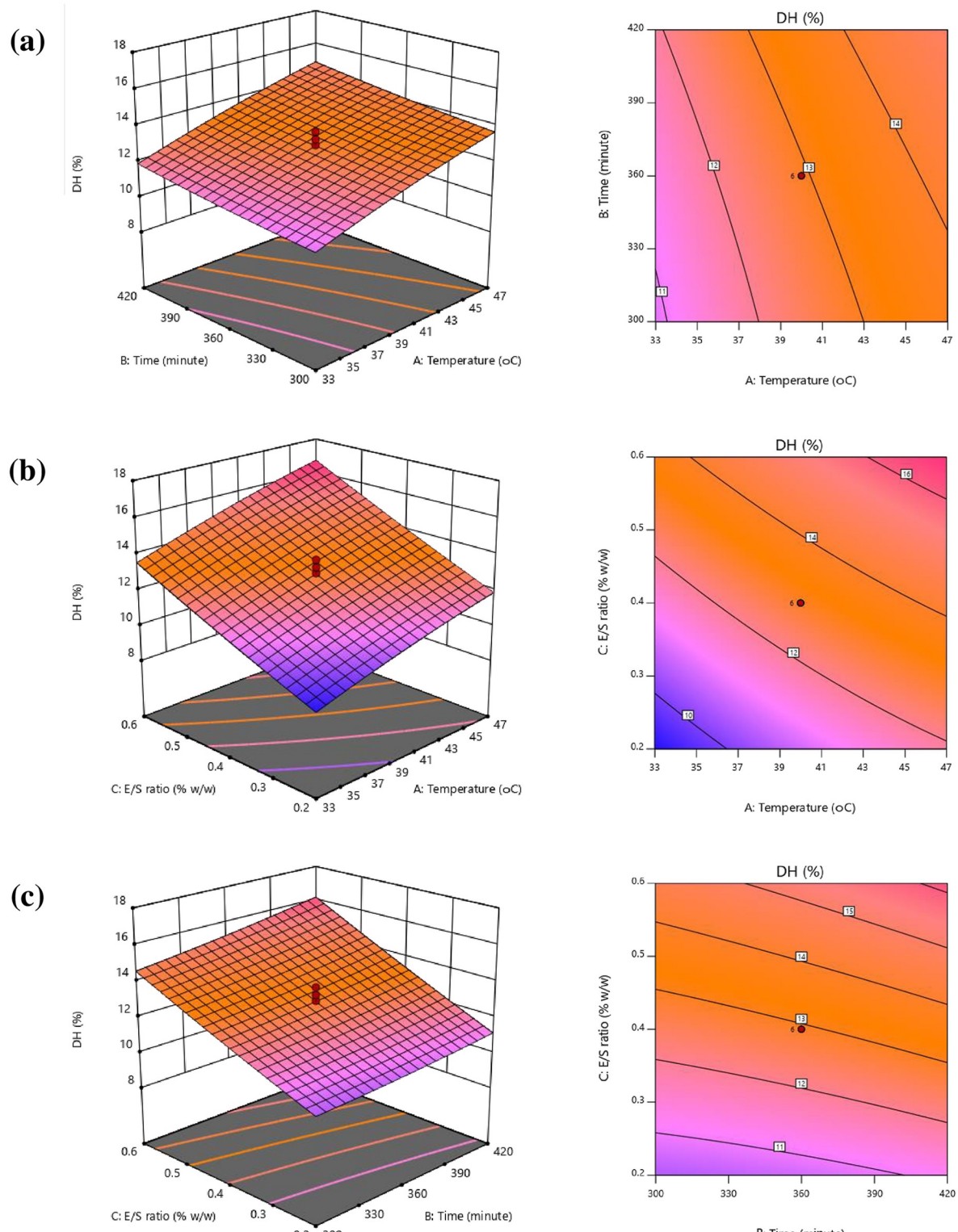

**Fig 2. Response surface and contour plots presenting the influence of interactions among independent variables upon the degree of hydrolysis (%DH).** (a) influence of temperature and time, (b) influence of temperature and E/S ratio, and (c) influence of time and E/S ratio.

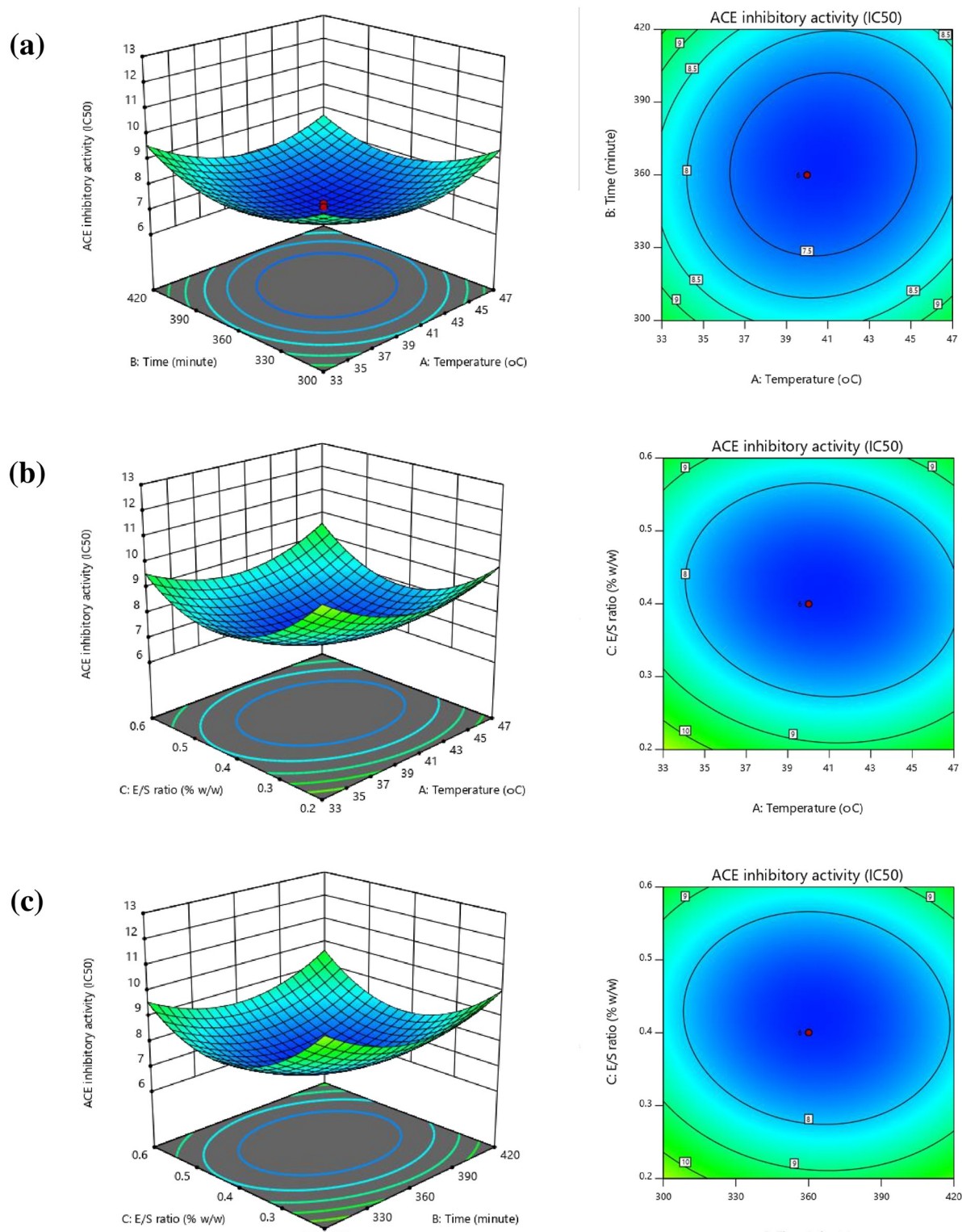

**Fig 3. Response surface and contour plots presenting the influence of interactions among independent variables upon ACE-inhibitory activity (IC$_{50}$; μg/mL).** (a) influence of temperature and time, (b) influence of temperature, and E/S ratio, and (c) influence of time and E/S ratio.

40.7˚C and the duration of hydrolysis is extended from 259.2 minutes to 363.92 minutes, the ACE-inhibitory activity of the SBPH also rises, but as the temperature continues to rise beyond 40.7˚C, the ACE inhibition begins to fall, while the time extends as far as 363.92 minutes. Meanwhile, Fig 3(b) shows how the temperature and E/S ratio combine to affect ACE-inhibitory activity, with the findings showing that ACE inhibition rises as the temperature and E/S ratio rise until an optimal value is reached, after which the ACE inhibition drops. Meanwhile, Fig 3(c) shows the quadratic influence of hydrolysis time and E/S ratio upon ACE inhibition, whereby the ACE-inhibitory activity increases as the two independent variables increase until reaching an optimal point, after which further increases in time and E/S ration see ACE-inhibitory activity start to decline.

Those hydrolysates which exhibited the greatest degree of hydrolysis at 17.56% produced $IC_{50}$ values measuring 10.44 μg/mL. In contrast, the DH value of 12.95% resulted in the lowest value for $IC_{50}$ of 6.87 μg/mL, as indicated in Table 3. It is clear from these findings that an increased degree of hydrolysis is not inherently associated with greater ACE-inhibitory activity or a lower value for $IC_{50}$. Dadzie *et al.* [37] concur, noting that in the case of vital wheat gluten hydrolysate, the greatest level of ACE inhibition resulted from relatively low DH values. However, Van der Ven *et al.* [43] reported, to the contrary, that whey protein hydrolysates generated using various pancreatic enzymes achieved high levels of ACE-inhibitory activity at high DH levels. It appears that the ability of peptide to inhibit ACE is mainly determined by their size and amino acid sequence [44]. For different fish hydrolysates, however, a connection has been found between low molecular weight peptides (less than 1 kDa) and high ACE-inhibitory activity [45]. As a result, the current study and others conclude that the amino acid structure and existence of free amino acids, in addition to peptide length, determine the ACE inhibitory activity of fish protein hydrolysates.

**Testing for validation.**   The optimal conditions for the processing were predicted using design expert statistical software based on the idea of a desirability index, which is an optimization approach involving multiple criteria for each response. The model produced a desirability value approaching 1, which implies that the proposed conditions would be the most appropriate to achieve the best DH values and ACE-inhibitory activity from the SBPH [37, 46]. The predicted optimal hydrolysis conditions to achieve the best value for DH were 47˚C, 420 minutes, and an E/S ratio of 0.6% (w/w). These conditions could produce a DH value of 17.68%. To optimize the ACE inhibition, the predicted optimal conditions were 40.7˚C, 363.92 minutes, and an E/S ratio of 0.42% (w/w). These conditions could produce and $IC_{50}$ value of 7.04 μg/mL (S1 Fig). These findings showed that the optimal conditions differed depending on whether the aim was to optimize DH or ACE inhibition, matching the results of Chen *et al.* [47], who also found that the optimal conditions for the DH value were not the same as the optimal conditions for the ACE-inhibitory activity.

In order to confirm the model validity, experiments were carried out in triplicate using the predicted optimal conditions. For DH, the experimental value achieved was a little lower than the prediction at 17.36 ± 0.69%, with an error value of 1.78%. For $IC_{50}$ the experimental value was a little higher than the prediction at 7.14 ± 0.05 μg/mL with an error value of 1.39%. Since the experiment produced results which were close to the prediction, it was concluded that the model had predicted the optimal conditions suitable for the production of ACE-inhibitory peptides. In addition, it was recorded that when the optimal conditions for ACE inhibition were employed, the resulting DH value was just 10.3%, which was substantially below the DH value of 17.36% which could be achieved in hydrolysis conditions were optimized for DH. Guo *et al.* [33] reported similar findings which give support to these results, noting that a lower DH value is associated with increased ACE inhibition in the context of the hydrolysates of whey protein concentrate when applying crude proteinases of *Lactobacillus helveticus* LB13.

## Ultrafiltration

In the production of bioactive peptides, the molecular weight of the protein hydrolysates is highly significant, and for this reason it is necessary to carry out the process of ultrafiltration in order to break the hydrolysate into different fractions of different molecular weights. This is usually done using different MWCO (molecular weight cut-off) membranes. This is a technique which can easily be scaled up to permit industrial-scale fractionation to produce ACE-inhibitory peptides [48]. This research selected the SBPH offering the greatest ACE-inhibitory potential to undergo further fractionation. The ultrafiltration process obtained five different fractions: MW > 10 kDa, 10–5 kDa, 5–3 kDa, 3–0.65 kDa, and < 0.65 kDa. Table 5 presents the ACE-inhibitory activity, which is clearly associated with the molecular weight. Those fractions whose molecular weights did not exceed 0.65 kDa provided the best ACE-inhibitory activity ($IC_{50}$ 0.093 ± 0.004 μg/mL) while the least effective ACE inhibition came from the fractions of molecular weight in the range of 5–10 kDa ($IC_{50}$ 42.640 ± 3.358 μg/mL). These findings suggest that the maximum ACE inhibition for the peptide can be increased if there is a decrease in the molecular weight of the hydrolysate fraction. Furthermore, it is important to note that the ACE-inhibition of peptides at different molecular weights will also depend upon the amino acid composition and sequence.

A number of researchers found similar results [49–51], observing that those peptides obtained via ultrafiltration which had low molecular weights had a tendency to exhibit greater levels of ACE-inhibitory activity than peptides of higher molecular weight. It has also been found that the intestines can readily absorb short-chain peptides of low molecular weight, and these are more beneficial in terms of suppressing hypertension than is the case for long-chain peptides [50]. This is because the short-chain peptides can resist gastrointestinal proteolysis and are more easily able to bind to the angiotensin-converting enzyme active site, thus wholly inhibiting the activity of the ACE enzyme, and consequently ensuring that angiotensin I is not converted to angiotensin II. It is therefore also reported that peptides with a low molecular weight are more effective than peptides of higher molecular weight in preventing ACE from facilitating angiotensin II production. The work of Pihlanto-Leppälä [52] supports these findings, having revealed that it is typical for 3–20 amino acid residues to be contained in a bioactive peptide, and that peptides with low molecular weights of less than 1 kDa exhibit greater ACE-inhibitory activity than those of higher molecular weight in the context of peptides obtained from protein hydrolysates of bovine whey. The study of Li *et al.* [53] investigating loach hydrolysates added further confirmation of this pattern, whereby a molecular weight below 2.5 kDa produced the greatest ACE inhibition, achieving an $IC_{50}$ value of 231.2 μg/mL,

**Table 5. ACE-inhibitory activity ($IC_{50}$) of the unfractionated trypsin hydrolysate and the five fractions obtained via ultrafiltration.**

| Molecular weight (kDa) | ACE-inhibitory activity ($IC_{50}$; μg/mL) |
|---|---|
| Unfractionated | 7.140 ± 0.051[b] |
| >10 | 21.556 ± 0.891[c] |
| 5–10 | 42.640 ± 3.358[d] |
| 3–5 | 2.078 ± 0.424[a] |
| 0.65–3 | 0.303 ± 0.025[a] |
| < 0.65 | 0.093 ± 0.004[a] |

All values are presented in the form of ACE-inhibitory activity mean ($IC_{50}$ value) ± Std. Deviation, with the tests performed in triplicate. When a letter in superscript is presented in the same column, this is indicative of a significant difference when applying Duncan's multiple range test (p < 0.05).

while the molecular weight above 10 kDa produced the least ACE inhibition. Gao *et al.* [33] carried out similar work studying the hydrolysis of cottonseed protein using papain before fractionation to obtain four fractions of differing molecular weights via ultrafiltration. The results showed that the fraction which had a molecular weight less than 5 kDa provided the most potent ACE inhibition, with a value for $IC_{50}$ of 0.792 mg/mL. Jung *et al.* [54] obtained similar results, also reporting that the fractionated yellowfin frame protein hydrolysates which had a molecular weight less than 5 kDa offered the greatest level of ACE-inhibitory activity, which showed some variation depending upon the distribution of the molecular mass. These findings clearly confirmed that following the ultrafiltration process, it can be expected that the peptides which have the lowest molecular weight will be the most effective in terms of ACE inhibition. For these reasons, this study selected the SBPH fraction with the lowest molecular weight to undergo additional purification using RP-HPLC. This was the fraction with the molecular weight less than 0.65 kDa.

## RP-HPLC

The use of RP-HPLC allowed the peptides to be separated on the basis of having differing levels of hydrophobicity, whereby hydrophobic molecules present in the mobile phase undergo adsorption to the hydrophobic stationary phase, thus enabling the elution of the hydrophilic molecules first of all. It is possible to lower the level of hydrophobic interaction in the solvent gradient by lowering the mobile phase polarity through raising the acetonitrile concentration. Consequently, this results in the desorption of the non-polar or hydrophobic molecules from the hydrophobic stationary phase thus permitting elution from the column. Taking the SBPH fraction with molecular weight lower than 0.65 kDa, which offered superior ACE inhibition resulted in heightened concentration through the use of a vacuum concentrator prior to purification through RP-HPLC using a Luna 5U C18 column with the gradient elution of acetonitrile (0–45%) for 50 minutes. Fig 4 presents the elution RP-HPLC profiles for each of the fractions along with details of the ACE-inhibitory activity. In Table 6, the ACE-inhibitory activities of the sixteen peptide peaks were individually determined and the results are presented. The $F_7$ fraction provided the greatest ACE-inhibitory activity among those fractions tested, offering an $IC_{50}$ value of 0.26 ± 0.02 μg/mL. Another group of fractions, $F_9$-$F_{16}$, which are derived from the mid-late phase of the RP-HPLC chromatogram, were also able to exhibit ACE inhibition, but for some of those fractions, namely $F_9$ and $F_{14}$-$F_{16}$, the ACE-inhibitory activity was lower than reported for other fractions. These findings concur with the results presented by Rui *et al.* [55], whose work indicated that higher levels of ACE-inhibitory activity are to be expected from peptides which contain higher quantities of hydrophobic amino acids and typically arise at the mid-late phase of the chromatogram.

In contrast, there was no ACE-inhibitory activity reported during the initial phases of the elution for $F_{1-6}$ and $F_8$. It was not possible to determine an $IC_{50}$ vale for these fractions due to the low protein concentration and the low level of ACE inhibition. From these findings, it can be inferred that the lack of ACE inhibition for these fractions could be explained by the fact that their peptide sequences contain higher levels of hydrophilic amino acid content, since this can block access to the ACE active site, thus eliminating the possibility of any inhibitory effects [56]. These are findings which concur with those of Yea *et al.* [57] who noted the importance of hydrophobicity as a factor affecting the ACE-inhibitory activity of the peptides, whereby those peptides which were highly hydrophilic or hydrophobic would not inhibit ACE. In this regard, it is important to consider the hydrophilic–hydrophobic ratio of a peptide when assessing ACE-inhibitory activity. In this case, the $F_7$ fractions were chosen to undergo analysis of the amino acid sequence in preparation for further testing.

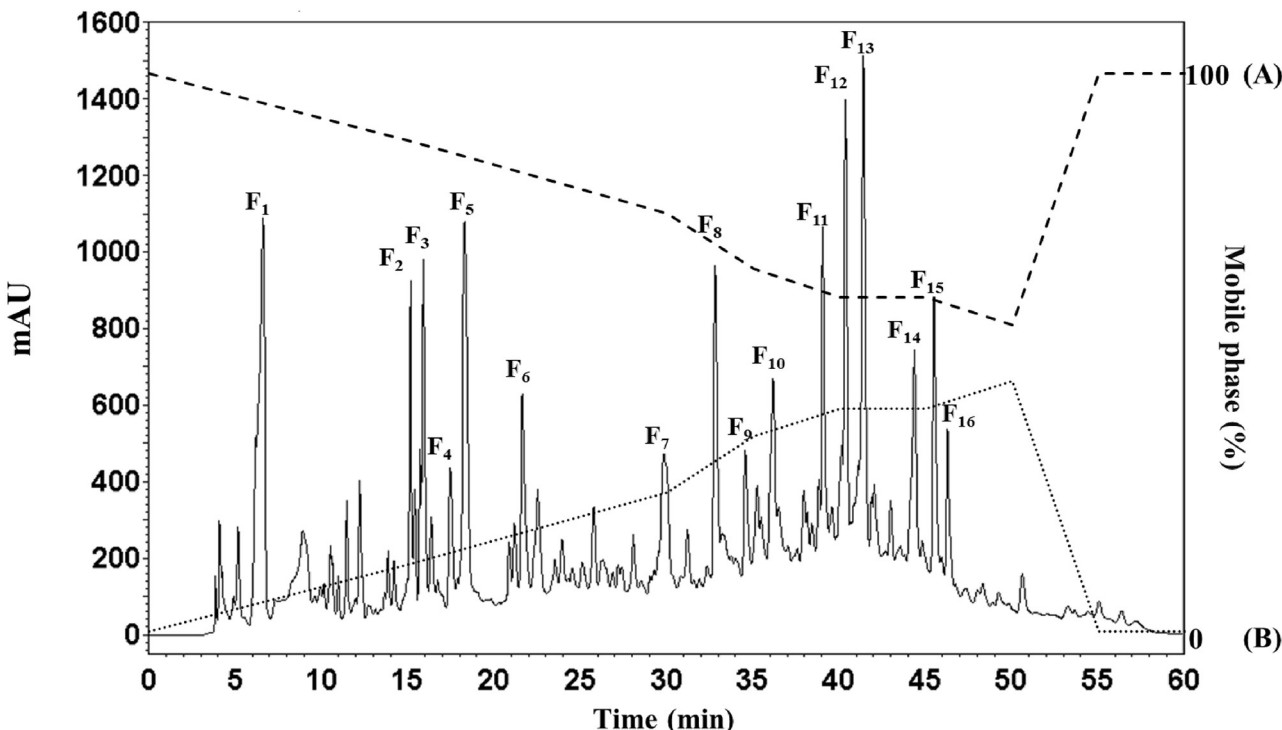

**Fig 4. The RP-HPLC chromatogram of the molecular weight cut-off < 0.65 kDa fraction of the trypsin hydrolysate obtained from salmon bones.**

**Table 6. ACE-inhibitory activity for each of the sixteen fractions obtained via RP-HPLC in terms of the $IC_{50}$ value (µg/mL).**

| Fractions | ACE-inhibitory activity ($IC_{50}$; µg/mL) |
|-----------|--------------------------------------------|
| $F_1$ | N.D. |
| $F_2$ | > 0.28 |
| $F_3$ | > 0.25 |
| $F_4$ | > 0.29 |
| $F_5$ | > 0.25 |
| $F_6$ | > 0.16 |
| $F_7$ | $0.26 \pm 0.02^a$ |
| $F_8$ | > 1.09 |
| $F_9$ | $1.60 \pm 0.15^d$ |
| $F_{10}$ | $1.12 \pm 0.05^{b, c}$ |
| $F_{11}$ | $0.34 \pm 0.01^a$ |
| $F_{12}$ | $0.88 \pm 0.01^b$ |
| $F_{13}$ | $0.97 \pm 0.02^{b, c}$ |
| $F_{14}$ | $1.22 \pm 0.11^c$ |
| $F_{15}$ | $1.70 \pm 0.34^d$ |
| $F_{16}$ | $1.22 \pm 0.20^c$ |

All values are presented in the form of ACE-inhibitory activity mean ($IC_{50}$ value) ± Std. Deviation, with the tests performed in triplicate. When a letter in superscript is presented in the same column, this is indicative of a significant difference when applying Duncan's multiple range test ($p < 0.05$).

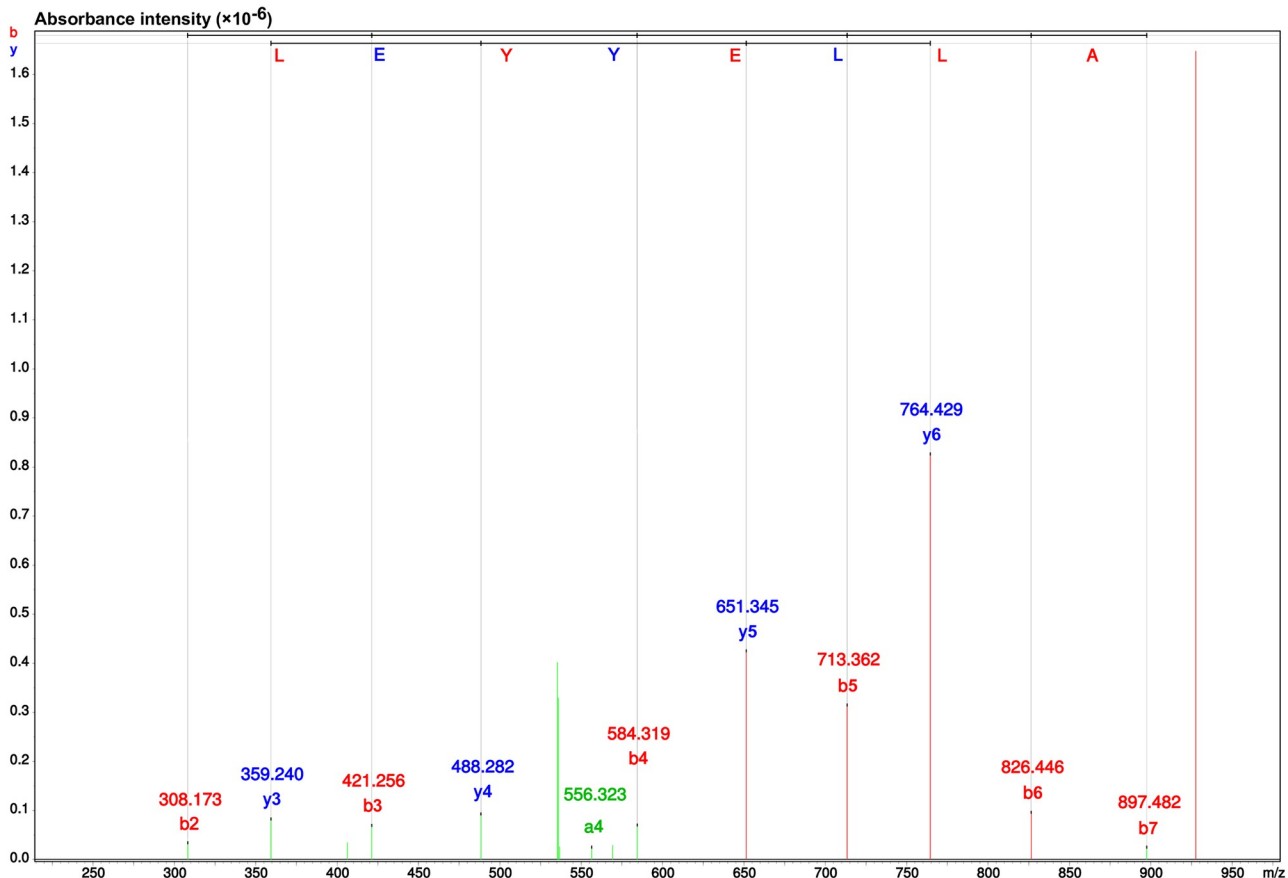

**Fig 5. MS/MS spectrum analysis and amino acid sequence details for the ACE-inhibitory FCLYELAR peptide from the F$_7$ fraction obtained via RP-HPLC.**

## Determination of the F$_7$ fraction amino acid sequence and *de novo* peptide sequencing

The F$_7$ fraction from RP-HPLC which offered the greatest ACE inhibition underwent further analysis to determine the amino acid sequences using both Nano-Liquid chromatography and a micrOTOF-Q IITM ESI-Qq-TOF mass spectrometer. The LC-MS/MS data allowed *de novo* sequencing based on further information obtained from the Mascot database and tested against the NCBI database. Fig 5 (S2 Fig) presents the findings from the mass spectrum analysis which show that the fraction F$_7$ peptide sequence comprised eight amino acids which were in the sequence: FCLYELAR (Phe-Cys-Leu-Tyr-Glu-Leu-Ala-Arg). Meanwhile, the molecular weight was shown to be 1,041.2 Da. The NCBI database along with the BLASTp program supported the identification of the protein in the case of the FCLYELAR sequence, with the findings revealing that the sequence was a 100% match with the protein of *Oncorhynchus mykiss*, and 85.71% match with an AT-rich interactive domain-containing protein 2-like isoform X1 of *Salmo salar*, and a 100% match with an ankyrin repeat domain-containing protein 55 of *S. salar*. The FCLYELAR sequence underwent chemical synthesis prior to estimation of its IC$_{50}$ value. The most potent activity was exhibited by the synthesized peptide FCLYELAR, recording an IC$_{50}$ value of 31.63 ± 0.15 μM. Evidence from the BIOPEP database, illustrated in Table 7, showed that the FCLYELAR peptide has ACE-inhibitory properties due to the ACE-inhibitory sequences LY, LA, AR, and YE. There have been no previous reports describing the

Table 7. The profiles for the properties of the FCLYELAR peptide.

| Peptide | Solubility in water[1] | Hydrophobicity (%) [2] | Toxicity profile (SVM score) [3] | ACE inhibitory sequences[4] | Sensory characteristics[5] |
|---------|----------------------|----------------------|--------------------------------|---------------------------|---------------------------|
| FCLYELAR | Poor | 50 | Non-Toxin (-0.51) | LY, LA, AR, and YE | EL (umami, bitter), LY (bitter), LA (bitter), AR (salt enhancer) |

[1]Data obtained from the Innovagen server concerning peptide solubility.

[2]Calculations were performed on the peptide property calculator (www.peptide2.com).

[3]Outcomes from peptide toxicity analysis.

[4,5]Data provided by the BIOPEP database.

purified FCLYELAR peptide, which can therefore be considered as a novel ACE-inhibitory peptide.

These results concur with reports suggesting that various peptides derived from marine life can inhibit ACE. For example, Phe-Leu was isolated from the hydrolysate of chum salmon muscle by Ono *et al.* [58]. When hydrolyzed using distilled water and 5% thermolysin, the reported value for $IC_{50}$ was 13.6 μM. Meanwhile, a similar study from Je *et al.* [59] showed that the peptide sequence Phe-Gly-Ala-Ser-Thr-Arg-Gly-Ala obtained from the protein hydroly-sates of Alaska pollack with pepsin produced a value for $IC_{50}$ 14.7 μM. In addition, peptides offering antihypertensive capabilities obtained from the hydrolysates of flounder muscles were purified by Ko *et al.* [60], who reported that the peptides Met-Glu-Val-Phe-Val-Pro (721.2 Da) and Val-Ser-Gln-Leu-Thr-Arg (703.4 Da) produced respective $IC_{50}$ values of 79 μM and 105 μM. The research of Sun *et al.* [61] determined the most potent amino sequences for ACE-inhibitory peptides derived from the marine macroalga, *Ulva intestinalis*, to be Phe-Gly-Met-Pro-Leu-Asp-Arg (834.41 Da) and Met-Glu-Leu-Val-Leu-Arg (759.43 Da), offering the respective $IC_{50}$ values of 219.35 μM and 236.85 μM. Meanwhile, a study conducted by Lee *et al.* [62] revealed that a peptide obtained from tuna frame protein offered ACE-inhibitory activity, and could be identified as Gly-Asp-Leu-Gly-Lys-Thr-Thr-Thr-Val-Ser-Asn-Trp-Ser-Pro-Pro-Lys-Try-Lys-Asp-Thr-Pro (2,482 Da). The $IC_{50}$ value in this case was 11.28 μM.

The extent to which a peptide exhibits ACE-inhibitory activity depends on certain aspects of the properties and structure, such as amino acid composition, chain length, molecular weight, charge, and hydrophobicity. Moreover, peptides which offer ACE inhibition are usually short peptides with 2–12 amino acid residues [63]. Some of these residues are of specific amino acid types and are found at the C-terminal end and/or N-terminal end. At the C-terminal end, there are three amino acid residues whose hydrophobicity properties are significantly affected when they are bound to ACE, since this contains hydrophobic amino acids [52, 64]. It is argued that the amino acid residues predominantly found at the penultimate position of the C-terminal end include aliphatic (Val, Ile, and Ala), basic (Arg), and aromatic (Tyr and Phe) types, while the ultimate position at the C-terminal end has predominantly aromatic (Trp, Tyr, and Phe), proline (Pro), and aliphatic (Ile, Ala, Leu, and Met) residues [65, 66]. It is sug-gested that it is these amino acid residues at those specific locations on the C-terminal end of the peptides which leads to improved ACE-inhibitory activity as a consequence of heightened binding with the ACE active site than other amino acids. Furthermore, the C-terminal is the location of the positively charged arginine (Arg) and lysine (Lys) which are understood to play a major role in the ACE inhibition provided by a number of different peptides [67].

On the basis of our own results, SBPH offered the novel ACE-inhibitory peptide, Phe-Cys-Leu-Tyr-Glu-Leu-Ala-Arg, whose hydrophobic amino acids had aliphatic side chains (Leu and Ala) at the penultimate position of the C-terminal, along with hydrophilic amino acid offering

a positive charge of arginine at the final position of the C-terminal end. These findings can explain the increased level of effectiveness in terms of ACE inhibition. Furthermore, negatively charged amino acids of glutamic acid (Glu) are also contained at the fifth position of the C-terminus. It is argued that negatively charged amino acids at the ACE active site are able to interact with $Zn^{2+}$ to lower the catalytic rate via the chelation of the vital zinc atoms which are critical if enzyme activity is to take place. Accordingly, it may be the case that negatively charged amino acids achieve potency in ACE inhibition [68]. Furthermore, the ACE-inhibitory peptides from the SBPH contain hydrophobic amino acids which have aromatic side chains comprising phenylalanine (Phe) at the N-terminal end along with tyrosine (Tyr) as the amino acid residue preferred at the middle position. It could be the case that ACE inhibition is significantly influenced by these amino acids. Therefore, the ACE-inhibitory ability of peptides could be affected by the amino acids which exist at any of the positions within the sequence, in addition to the well-understood C-terminal tripeptide sequence.

### *In silico* peptide property analysis and prediction of sensory analysis

Table 7 presents the predictions for the properties of peptides and their sensory analysis. According to the date drawn from the Innovagen server, it would be anticipated that the water solubility of the FCLYELAR peptide should be poor. In contrast, however, our research showed good solubility when the peptides were immersed in ultrapure water at concentrations not exceeding 5.0 mg/mL. In the analysis of peptide hydrophobicity, it was shown that FCLYELAR was 50% hydrophobic due to the presence of hydrophobic amino acid residues comprising Phe, Cys, Leu, Tyr, and Ala. From analysis via ToxinPred, it can be predicted that ACE-inhibitory peptides are non-toxic, having recorded an SVM score of -0.51. Moreover, sensory analysis revealed that the taste of FCLYELAR is predominantly bitter, with a little umami and slightly salty. The dominant bitter taste may be the consequence of the amino acids which have tyrosine, leucine, phenylalanine, and Arg at the peptide chain C- or N-terminals [69]. The FCLYELAR peptide also contains glutamic acid, which could be responsible for the umami aspect of the taste, as earlier reports have associated glutamic acid with peptides that exhibit an umami taste [70].

### Inhibitory kinetics of ACE inhibitory peptides

In order to determine the inhibition patterns of the FCLYELAR peptides, this study made use of Lineweaver-Burk plots. Different concentrations of HHL (0.5, 1, 3, 5, and 7 mM) underwent incubation in ACE solution both with and without (the control) varying concentrations of the FCLYELAR peptides (0.05, 0.1 and 0.2 mM). Fig 6 (S4 Table) presents the results, which show an uncompetitive inhibition pattern in which lines close to parallel were generated. The x- and y-intercepts of these lines occurred at different points as the peptide concentration was increased. On the basis of the plots, it appears that the FCLYELAR peptide obtained from SBPH is able to bind solely to the enzyme-substrate complex but is not capable of binding to the free enzyme. However, the ACE inhibitor is never in competition with the HHL in attempting to bind at the same active site as ACE, and therefore it is not possible to strengthen the inhibition through an increase in the concentration of either the substrate or inhibitor. The peptides may therefore bind to other ACE sites which are not the same as the substrate (HHL) binding site, and this can lead to an alteration in the enzyme conformation, causing reduced ACE activity. Accordingly, there is modification of Km and Vmax but in steady-state circumstances, there is a kinetic pattern which appears.

Table 8 presents the kinetics parameters, whereby it can be observed that there is a decline in the Michaelis–Menten constant (Km) and Vmax as the concentration of the FCLYELAR

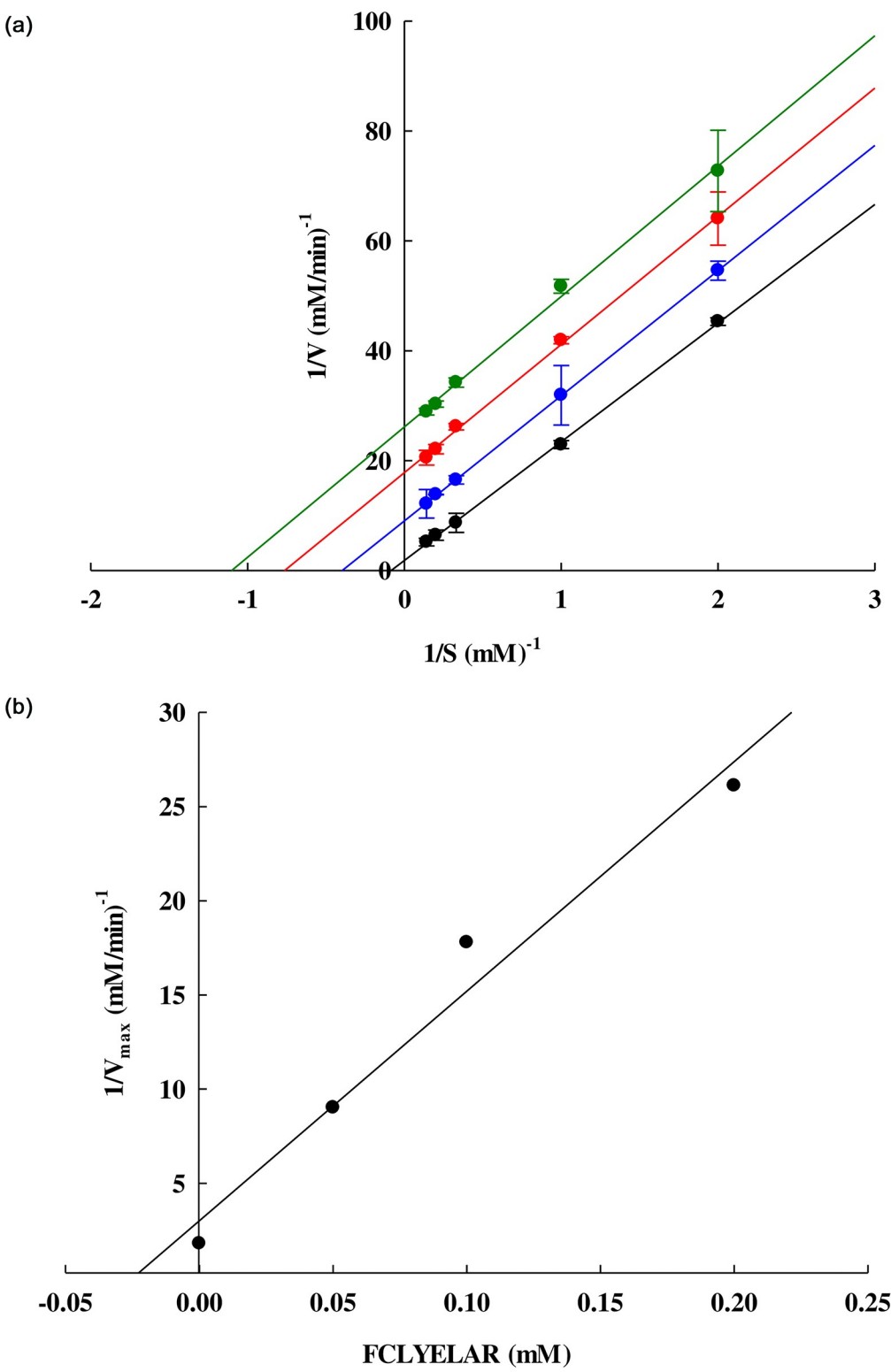

**Fig 6.** (a) The Lineweaver-Burk plot illustrating the influence of FCLYELAR uncompetitive inhibition upon ACE. The ACE-inhibitory activities were assessed when various concentrations of FCLYELAR peptide were present or absent (●, control; ●, 0.05 mM; ●, 0.1 mM; ●, 0.2 mM). 1/V and 1/S are the respective reciprocals of the velocity and substrate. Each of the points is presented in the form of mean value ± Std. Deviation with tests performed in triplicate, and (b) the secondary plots showed the determination of the inhibitor constant ($K_i$) in the case of the uncompetitive inhibition by FCLYELAR.

Table 8. The kinetics parameters for reactions undergoing catalysis via ACE at differing peptide concentrations.

| Kinetics Parameters | control | FCLYELAR (mM) | | |
|---|---|---|---|---|
| | | 0.05 | 0.1 | 0.2 |
| $K_m$ (mM) | 12.0219 | 2.5293 | 1.3130 | 0.9099 |
| $V_{max}$ (mM/min) | 0.5558 | 0.1109 | 0.0562 | 0.0383 |
| $K_i$ (mM) | | 0.0248 | | |

The results are findings are expressed in the form of mean ± SEM (n = 3).

peptide increases. When the peptide was not present at all, the value of Km 12.0219 mM. When the peptide was gradually introduced in concentrations measuring 0.05, 0.1, and 0.2 mM, the respective Km values were 2.5293, 1.3130, and 0.9099 mM. For Vmax, when the peptide was not present, the value was 0.5558 mM/min, and when the peptide was gradually introduced (0.05, 0.1 and 0.2 mM), the respective Vmax values were 0.1109, 0.0562 and 0.0383 (mM/min). Ki represented the inhibitor constant, which is a measure of the binding strength as the peptide forms bonds with ACE. A higher affinity is associated with a lower Ki value. In the case of the FCLYELAR peptide, Ki measured a constant 0.0248 mM as shown in Fig 6.

A majority of the peptides investigated have been shown to act as competitive inhibitors for ACE. Trp-Pro-Met-Gly-Phe (636.75 Da) is one such example, obtained from the trypsin hydrolysates of *Cyclina sinensis* [71], along with Val-Val-Ser-Leu-Ser-Ile-Pro-Arg (869.53 Da) which comes from pigeon pea seeds [72]. There have been some ACE-inhibitory peptides reported to carry out their ACE-inhibitory activity through an uncompetitive pattern, including Ala-Trp, Phe-Tyr, and Ile-Trp which are derived from wakame hydrolysates of (*Undaria pinnatifida*) using protease [73], and also Phe-Glu-Ser-Asn-Phe-Asn-Thr-Gln-Ala-Thr-Asn-Arg which has a molecular weight of 1428.6 Da and is obtained from the white lysozyme (HEWL) hydrolysates of hens' eggs (Ki = 0.082 mg/mL) [74]. Another example is Tyr-Leu-Tyr-Glu-Ile-Ala-Arg-Arg which is derived from the tryptic hydrolysate found in human plasma [75]. Girgih *et al*. [76] made similar reports, observing that uncompetitive inhibition was displayed by unfractionated hydrolysate peptides (CPH) and the CF3 fraction, which produced respective Ki values of 0.263 and 0.394 mg/mL. In the work of Cian *et al*. [77], it was found that wheat gluten hydrolysate peptides (GHE) performed uncompetitive inhibition, whereas Jao *et al*. [78] claimed that peptides acting as competitive or uncompetitive inhibitors have inhibition sites which are not clearly specified, and therefore there remains a lack of clarity over the exact mechanism by which ACE-inhibitory peptides are able to perform their inhibition activity. For this reason, it will be necessary to conduct further research into peptide structures and inhibition modes to better understand the relationship.

## Molecular docking of peptides with ACE

Molecular docking simulations were carried out using the GOLD program and Discovery Studio 2019 in order to model the interaction between the ACE structure and the peptide functions. In particular, this study sought to predict how the FCLYELAR peptide would interact with the ACE structure. Previous research has shown that the ACE active site comprises three pockets, known as S1, S2', and S1' [79, 80]. Residues in the S1 pocket include Ala354, Glu384, and Tyr523; S2' contains Gln281, His353, Lys511, His513, and Tyr520, while S1' contains only Glu162. It was also noted previously that in inhibiting ACE activity, the way the ACE inhibitors and the $Zn^{2+}$ interact at the ACE active site is of particular importance as the zinc ion at the active site coordinates with the ACE residues His383, His387, and Glu411 [81].

**Table 9. Interactions between the ACE residues and the FCLYELAR peptide.**

| FCLYELAR | ACE residues | Category (Types) | From Chemistry | To Chemistry | Distance (Å) |
|---|---|---|---|---|---|
| F: Phe1 | Glu337 | Electrostatic (Attractive Charge) | Phe1:N (Positive) | Glu337:OE2 (Negative) | 4.2136 |
| | Thr243 | Conventional Hydrogen Bond | Phe1:HT3 (H-Donor) | Thr243:OG1 (H-Acceptor) | 1.9759 |
| | Gln242 | Conventional Hydrogen Bond | Gln242:HE21 (H-Donor) | Phe1:O (H-Acceptor) | 2.6485 |
| | Val340 | Hydrophobic (Pi-Alkyl) | Phe1 (Pi-Orbitals) | Val340 (Alkyl) | 4.9222 |
| C: Cys2 | His314 | Conventional Hydrogen Bond | His314:HD1 (H-Donor) | Cys2:O (H-Acceptor) | 2.6499 |
| Y: Tyr4 | His314 | Conventional Hydrogen Bond | Tyr4:HN (H-Donor) | His314:NE2 (H-Acceptor) | 2.0228 |
| | His344 | Conventional Hydrogen Bond | His344:HE2 (H-Donor) | Tyr4:O (H-Acceptor) | 2.5424 |
| | His344 | Hydrophobic (Pi-Pi T-shaped) | His344 (Pi-Orbitals) | Tyr4 (Pi-Orbitals) | 5.7379 |
| | Tyr480 | Conventional Hydrogen Bond | Tyr480:HH (H-Donor) | Tyr4:O (H-Acceptor) | 2.2704 |
| | Tyr480 | Hydrophobic (Pi-Pi Stacked) | Tyr4 (Pi-Orbitals) | Tyr480 (Pi-Orbitals) | 4.8833 |
| | Zn576 | Metal-Acceptor | Zn576:Zn (Metal) | Tyr4:O (H-Acceptor) | 2.5894 |
| E: Glu5 | Ala315 | Conventional Hydrogen Bond | Glu5:HN (H-Donor) | Ala315:O (H-Acceptor) | 1.6987 |
| | His348 | Conventional Hydrogen Bond | His348:HE2 (H-Donor) | Glu5:OE1 (H-Acceptor) | 2.7819 |
| | Tyr480 | Conventional Hydrogen Bond | Tyr480:HH (H-Donor) | Glu5:O (H-Acceptor) | 2.8036 |
| | Tyr480 | Conventional Hydrogen Bond | Tyr480:HH (H-Donor) | Glu5:OE1 (H-Acceptor) | 2.6922 |
| | Zn576 | Metal-Acceptor | Zn576:Zn (Metal) | Glu5:OE1 (H-Acceptor) | 2.5731 |
| | Ala315 | Unfavorable Bump;Carbon Hydrogen Bond | Glu5:HA (Steric;H-Donor) | Ala315:O (Steric;H-Acceptor) | 1.5621 |
| A: Ala7 | Arg479 | Conventional Hydrogen Bond | Arg479:HH11 (H-Donor) | Ala7:O (H-Acceptor) | 2.2827 |
| R: Arg8 | Glu372 | Electrostatic (Attractive Charge) | Arg8:NE (Positive) | Glu372:OE2 (Negative) | 5.2081 |
| | His371 | Electrostatic (Pi-Cation) | Arg8:NE (Positive) | His371 (Pi-Orbitals) | 3.9520 |
| | Arg479 | Electrostatic (Attractive Charge) | Arg479:NH2 (Positive) | Arg8:O (Negative) | 4.3193 |
| | Arg479 | Conventional Hydrogen Bond | Arg479:HH11 (H-Donor) | Arg8:OXT (H-Acceptor) | 2.0674 |
| | Arg479 | Conventional Hydrogen Bond | Arg479:HH21 (H-Donor) | Arg8:OXT (H-Acceptor) | 2.1053 |
| | Arg479 | Unfavorable Positive-Positive | Arg8:NE (Positive) | Arg479:NH2 (Positive) | 5.4535 |

In this study, the evaluation of the molecular docking of the FCLYELAR peptide at the ligand A5 position of the ACE molecule revealed the highest-ranking docking position offered a PLP fitness score of 137.1516 (Ref. RMSD: 2.7754). The FCLYELAR peptide, however, forms no bonds with the subsite pockets, S1, S2', and S1', and is not found at the ACE active site. It can this be inferred that the FCLYELAR peptide is not involved in competition with HHL in order to bind at the same ACE active site. The findings lead to the conclusion that the interaction involving the FCLYELAR peptide and the ACE residues is uncompetitive, thus matching out own findings from the Lineweaver-Burk plot concerning ACE inhibition patterns. Further support for this inference comes from Ni *et al.* [82], who determined that the subsite ACE pockets were not occupied by TPTQQS, despite the significant role played by TPTQQS in ACE inhibition, thus suggesting that there had been a change in the ACE active site. Table 9 shows the outcomes of the docking results concerning the interaction between the FCLYELAR peptide and ACE, while Fig 7(a) illustrates a 3D structural prediction for the peptide-ACE complex. Fig 7(b) and 7(c) respectively portray the 3D and 2D descriptions of the anticipated interactions taking place between the synthesized FCLYELAR peptide and the ACE residues.

The mechanism by which the bonds between the FCLYELAR peptide and the ACE residues arose involved a complex framework of 13 conventional hydrogen bonds, four electrostatic bonds, three hydrophobic bonds, two metal-acceptor bonds, and finally two unfavorable bonds, all of which are presented in Table 9. Some reports have argued that it might be the interaction of the hydrogen bonds taking place between peptides and ACE active or non-active sites which is responsible for ACE inhibition, but hydrophobic and electrostatic interactions

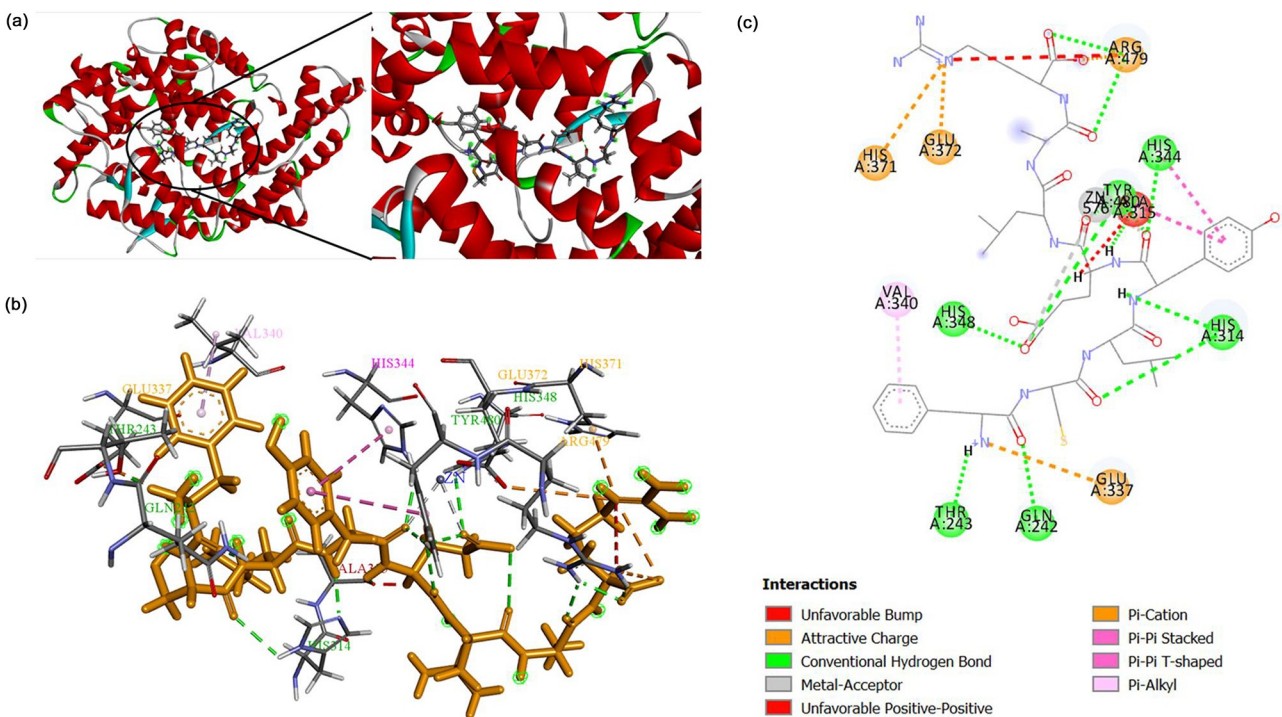

**Fig 7. Simulations of molecular docking for FCLYELAR binding to ACE (PDB: 1O8A).** (a) Broad overview and narrow localized perspective for the 3D structure prediction of the peptide-ACE complex; (b) Anticipated interactions involving the synthesized FCLYELAR peptides and ACE residues (Images produced using Discovery Studio Visualizer v19.1.0 software), (c) 2D representation of the anticipated interactions involving the synthesized FCLYELAR peptides and ACE molecules (Images produced using Discovery Studio Visualizer v19.1.0 software).

are also known to be important in inhibiting ACE activity [83]. In addition, the peptide sequence contains Tyr4 and Glu5 residues which can interact with ACE via metal-acceptor bonds (Zn576) which also play a key part in ACE inhibition. Meanwhile, it has also been found that negatively charged amino acids (Glu) at the ACE active site are able to bind to zinc ions, which once again would lead to ACE inhibition [68].

## Conclusion

The results of this work demonstrated that salmon bone has a high potential for usage as a material for isolating and characterizing ACE-inhibitory peptides via the trypsin hydrolysis process. Protein hydrolysates were prepared using RSM in combination with CCD to optimize the conditions for hydrolysis and achieve the best performance in terms of ACE inhibition and degree of hydrolysis. The greatest potency for ACE inhibition was achieved by setting the temperature to 40.7˚C, the time to 363.92 minutes, and the E/S ratio to 0.42% (w/w), thereby producing an $IC_{50}$ value of 7.14 μg/ml. Following that, the salmon bone hydrolysate was ultrafiltered and purified using RP-HPLC. Q-TOF-LC-MS/MS was used to identify the resulting peptide. According to the findings, fraction F7 has the best ACE-inhibitory potency. The ACE inhibitory peptide sequence was FCLYELAR, with a molecular weight of 1041.2 Da. According to the inhibitory patterns displayed in the Lineweaver-Burk plots, the synthesized FCLYELAR peptide was an uncompetitive inhibitor of ACE, with the lowest $IC_{50}$ value of 31.63 μM. Through molecular docking simulations of the predicted interactions between the synthesized peptides and ACE residues, it was discovered that the FCLYELAR peptide can connect to the non-active site of ACE, primarily via hydrogen bonds. In conclusion,

ACE-inhibitory peptides produced from salmon bone hydrolysates have shown the potential to efficiently block ACE activity in vitro and may thus play a role in the development of future medications that may be used to address the problem of hypertension. However, in vivo studies to determine the extent to which these peptides are absorbed in the gastrointestinal tract, as well as details about their cytotoxicity and allergenicity, will be required in order to safely develop beneficial food supplements or pharmaceutical products for human consumption.

## Supporting information

**S1 Fig.** The predicted DH and the desirability value for optimum selected conditions of different independent variables for hydrolysis, as shown in (a). The predicted ACE-inhibitory activity (IC50; μg/mL) and the desirability value for optimum selected conditions of different independent variables for hydrolysis, as shown in (b).
(DOCX)

**S2 Fig. MS/MS spectrum analysis and amino acid sequence details for the ACE-inhibitory FCLYELAR peptide from the F7 fraction obtained via RP-HPLC.**
(PDF)

**S1 Table. Preliminary assessment.**
(DOCX)

**S2 Table. Experimental design matrix of CCD and corresponding results (degree of hydrolysis).**
(DOCX)

**S3 Table. Experimental design matrix of CCD and corresponding results (ACE-inhibitory activity).**
(DOCX)

**S4 Table. Kinetic study.**
(DOCX)

## Acknowledgments

The authors were grateful to Dr. Robert Douglas John Butcher for reviewing this manuscript.

## Author Contributions

**Conceptualization:** Aphichart Karnchanatat.

**Data curation:** Thanakrit Kaewsahnguan, Sajee Noitang.

**Formal analysis:** Aphichart Karnchanatat.

**Funding acquisition:** Aphichart Karnchanatat.

**Investigation:** Aphichart Karnchanatat.

**Methodology:** Thanakrit Kaewsahnguan, Sajee Noitang, Papassara Sangtanoo, Piroonporn Srimongkol, Tanatorn Saisavoey.

**Project administration:** Aphichart Karnchanatat.

**Resources:** Thanakrit Kaewsahnguan, Sajee Noitang, Papassara Sangtanoo, Piroonporn Srimongkol, Tanatorn Saisavoey.

**Software:** Onrapak Reamtong, Kiattawee Choowongkomon.

**Validation:** Thanakrit Kaewsahnguan, Sajee Noitang.

**Visualization:** Thanakrit Kaewsahnguan, Aphichart Karnchanatat.

**Writing – original draft:** Thanakrit Kaewsahnguan, Aphichart Karnchanatat.

**Writing – review & editing:** Thanakrit Kaewsahnguan, Aphichart Karnchanatat.

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
