## [Decision Letter · Decision Letter 0]

22 Mar 2021

PONE-D-21-03572

A novel angiotensin I-converting enzyme inhibitory peptide derived from the trypsin hydrolysates of salmon bone proteins

PLOS ONE

Dear Dr. Karnchanatat,

Thank you for submitting your manuscript to PLOS ONE. After careful consideration, we feel that it has merit but does not fully meet PLOS ONE’s publication criteria as it currently stands. Therefore, we invite you to submit a revised version of the manuscript that addresses the points raised during the review process.

We look forward to receiving your revised manuscript.

Kind regards,

Shashi Kant Bhatia

Academic Editor

PLOS ONE

3.Thank you for stating the following in the Acknowledgments Section of your manuscript:

"Furthermore, the authors would also like to make clear their

12 gratitude for the financial support provided by the Thailand Science Research and

13 Innovation (TSRI) Fund (CU_FRB640001_01_61_1), and the Center of Excellence on

14 Medical Biotechnology (CEMB), S&T Postgraduate Education and Research

15 Development Office (PERDO), Office of Higher Education Commission (OHEC),

16 Thailand (SN-63-009-01). The assistance provided by the aforementioned bodies has made

17 this research possible."

4. Please upload a copy of Figure 8, to which you refer in your text on page 32. If the figure is no longer to be included as part of the submission please remove all reference to it within the text.

Reviewers' comments:

Reviewer's Responses to Questions

**Comments to the Author**

1. Is the manuscript technically sound, and do the data support the conclusions?

Reviewer #1: Yes

Reviewer #2: Yes

2. Has the statistical analysis been performed appropriately and rigorously? 

Reviewer #1: Yes

Reviewer #2: Yes

3. Have the authors made all data underlying the findings in their manuscript fully available?

Reviewer #1: Yes

Reviewer #2: Yes

4. Is the manuscript presented in an intelligible fashion and written in standard English?

Reviewer #1: Yes

Reviewer #2: No

5. Review Comments to the Author

Reviewer #1: Authors have presented a process of enzymatic hydrolysis of salmon bone with trypsin for production of ACE inhibitory peptides. Authors should correct some of the mistakes in the English of the manuscript. Authors should discuss the similar studies (e.g., Fu-Yuan Cheng et al., 2009; Fatemeh Mahmoodani et al., 2014; Isabel Rodríguez Amado et al., 2014, etc.) on bone hydrolysis for peptide production, and difference from the present findings should be emphasized.

Conclusion should be more concise.

Reviewer #2: Journal: PLOS ONE

Manuscript ID PONE-D-21-03572

Title: A novel angiotensin I-converting enzyme inhibitory peptide derived from the trypsin hydrolysates of salmon bone proteins

General Comments

Thanakrit, et al., extracted a novel angiotensin I-converting enzyme inhibitory peptide from the trypsin hydrolysates of salmon bone proteins. The objectives of manuscript are clear but needs extensive details on its background of the research and goal. As well, some points in the results and discussion part are incomprehensible and require an in-depth explanation.

Authors should answer following questions before publication

1. The manuscript title not providing informative details on the study carried out

2. English polishing is required before publishing this manuscript

3. Introduction is too vague and descriptive; it must be concise and informative considering aims and objectives of the study

4. For proximate analysis provide AOAC updated methods from 2000 onwards.

5. In every section the authors provided too much descriptive information on each study point, they should provide specific and concrete details on each parameter studied.

6. Some data must provide as supplementary files

---

## [Author Response · Author response to Decision Letter 0]

14 May 2021

List of changes or point rebuttals

We would like to thank the reviewer for careful and thorough reading of this manuscript and for the thoughtful comments and constructive suggestions, which help to improve the quality of this manuscript. Our response follows (the reviewer’s comments are in yellow highlights).

- The suggested correction has been made. 

- The suggested correction has been made. 

"Furthermore, the authors would also like to make clear their

12 gratitude for the financial support provided by the Thailand Science Research and

13 Innovation (TSRI) Fund (CU_FRB640001_01_61_1), and the Center of Excellence on

14 Medical Biotechnology (CEMB), S&T Postgraduate Education and Research

15 Development Office (PERDO), Office of Higher Education Commission (OHEC),

16 Thailand (SN-63-009-01). The assistance provided by the aforementioned bodies has made

17 this research possible."

- The suggested correction has been made. 

4. Please upload a copy of Figure 8, to which you refer in your text on page 32. If the figure is no longer to be included as part of the submission please remove all reference to it within the text.

- The suggested correction has been made. 

Reviewers' comments:

Reviewer's Responses to Questions

Comments to the Author

1. Is the manuscript technically sound, and do the data support the conclusions?

Reviewer #1: Yes

Reviewer #2: Yes

2. Has the statistical analysis been performed appropriately and rigorously?

Reviewer #1: Yes

Reviewer #2: Yes

3. Have the authors made all data underlying the findings in their manuscript fully available?

Reviewer #1: Yes

Reviewer #2: Yes

4. Is the manuscript presented in an intelligible fashion and written in standard English?

Reviewer #1: Yes

Reviewer #2: No

5. Review Comments to the Author

Reviewer #1: Authors have presented a process of enzymatic hydrolysis of salmon bone with trypsin for production of ACE inhibitory peptides. Authors should correct some of the mistakes in the English of the manuscript. Authors should discuss the similar studies (e.g., Fu-Yuan Cheng et al., 2009; Fatemeh Mahmoodani et al., 2014; Isabel Rodríguez Amado et al., 2014, etc.) on bone hydrolysis for peptide production, and difference from the present findings should be emphasized.

Conclusion should be more concise.

- Additional discussion is in the response surface and contour plots section

Reviewer #2: Journal: PLOS ONE

Manuscript ID PONE-D-21-03572

Title: A novel angiotensin I-converting enzyme inhibitory peptide derived from the trypsin hydrolysates of salmon bone proteins

General Comments

Thanakrit, et al., extracted a novel angiotensin I-converting enzyme inhibitory peptide from the trypsin hydrolysates of salmon bone proteins. The objectives of manuscript are clear but needs extensive details on its background of the research and goal. As well, some points in the results and discussion part are incomprehensible and require an in-depth explanation.

Authors should answer following questions before publication

1. The manuscript title not providing informative details on the study carried out.

- The manuscript title is appropriate for this article and representative of the content. It also enables readers outside the subject to understand the purpose.

2. English polishing is required before publishing this manuscript.

- Dr. Robert Douglas John Butcher for reviewing this manuscript.

3. Introduction is too vague and descriptive; it must be concise and informative considering aims and objectives of the study.

- The suggested correction has been made. 

4. For proximate analysis provide AOAC updated methods from 2000 onwards.

- The AOAC methods were updated in 2005.

5. In every section the authors provided too much descriptive information on each study point, they should provide specific and concrete details on each parameter studied.

- The suggested correction has been made. 

6. Some data must provide as supplementary files.

- The suggested correction has been made.

---

## [Decision Letter · Decision Letter 1]

29 Jul 2021

PONE-D-21-03572R1

A novel angiotensin I-converting enzyme inhibitory peptide derived from the trypsin hydrolysates of salmon bone proteins

PLOS ONE

Dear Dr. Karnchanatat,

Thank you for submitting your manuscript to PLOS ONE. After careful consideration, we feel that it has merit but does not fully meet PLOS ONE’s publication criteria as it currently stands. Therefore, we invite you to submit a revised version of the manuscript that addresses the points raised during the review process.

We look forward to receiving your revised manuscript.

Kind regards,

Shashi Kant Bhatia

Academic Editor

PLOS ONE

Reviewers' comments:

Reviewer's Responses to Questions

**Comments to the Author**

Reviewer #1: Authors need to address the reviewers' comments more appropriately. Authors are advised to answer the comments describing the changes made in the manuscript. I can see in response to some of the comments any changes have not been done.

Some examples are:

2.English polishing is required before publishing this manuscript.

-Dr. Robert Douglas John Butcher for reviewing this manuscript.

3.Introduction is too vague and descriptive; it must be concise and informative

considering aims and objectives of the study.

-The suggested correction has been made.

---

## [Author Response · Author response to Decision Letter 1]

31 Jul 2021

List of changes or point rebuttals

We would like to thank the reviewers for their careful and thorough reading of this manuscript and for their thoughtful comments and constructive suggestions, which helped to improve the quality of this manuscript. Our response follows (the reviewer’s comments are in yellow highlights).

- The suggested correction has been made. 

- The suggested correction has been made. 

"Furthermore, the authors would also like to make clear their

12 gratitude for the financial support provided by the Thailand Science Research and

13 Innovation (TSRI) Fund (CU_FRB640001_01_61_1), and the Center of Excellence on

14 Medical Biotechnology (CEMB), S&T Postgraduate Education and Research

15 Development Office (PERDO), Office of Higher Education Commission (OHEC),

16 Thailand (SN-63-009-01). The assistance provided by the aforementioned bodies has made

17 this research possible."

- The suggested correction has been made. 

4. Please upload a copy of Figure 8, to which you refer in your text on page 32. If the figure is no longer to be included as part of the submission please remove all reference to it within the text.

- The suggested correction has been made. 

Reviewers' comments:

Reviewer's Responses to Questions

Comments to the Author

1. Is the manuscript technically sound, and do the data support the conclusions?

Reviewer #1: Yes

Reviewer #2: Yes

2. Has the statistical analysis been performed appropriately and rigorously?

Reviewer #1: Yes

Reviewer #2: Yes

3. Have the authors made all data underlying the findings in their manuscript fully available?

Reviewer #1: Yes

Reviewer #2: Yes

4. Is the manuscript presented in an intelligible fashion and written in standard English?

Reviewer #1: Yes

Reviewer #2: No

5. Review Comments to the Author

Reviewer #1: Authors have presented a process of enzymatic hydrolysis of salmon bone with trypsin for production of ACE inhibitory peptides. 

-Authors should correct some of the mistakes in the English of the manuscript. 

The current version has been carefully revised to improve the grammar and readability by Chulalongkorn University Language Institute.

-Authors should discuss the similar studies (e.g., Fu-Yuan Cheng et al., 2009; Fatemeh Mahmoodani et al., 2014; Isabel Rodríguez Amado et al., 2014, etc.) on bone hydrolysis for peptide production, and difference from the present findings should be emphasized. 

In this work, similar studies such as the work by Rasli H. and Sarbon N., 2018; Chen J., et al., 2009.; Zhu Z., et al., 2010 have been discussion in the optimization of salmon bone hydrolysis section.

- Conclusion should be more concise.

The current version has been modified.

Reviewer #2: Journal: PLOS ONE

Manuscript ID PONE-D-21-03572

Title: A novel angiotensin I-converting enzyme inhibitory peptide derived from the trypsin hydrolysates of salmon bone proteins

General Comments

Thanakrit, et al., extracted a novel angiotensin I-converting enzyme inhibitory peptide from the trypsin hydrolysates of salmon bone proteins. The objectives of manuscript are clear but needs extensive details on its background of the research and goal. As well, some points in the results and discussion part are incomprehensible and require an in-depth explanation.

Authors should answer following questions before publication

1. The manuscript title not providing informative details on the study carried out.

- The manuscript title is appropriate for this article and representative of the content. It also enables readers outside the subject to understand the purpose.

2. English polishing is required before publishing this manuscript.

- The current version has been carefully revised to improve the grammar and readability by Chulalongkorn University Language Institute.

3. Introduction is too vague and descriptive; it must be concise and informative considering aims and objectives of the study.

- The current version has been modified (in yellow highlights).

4. For proximate analysis provide AOAC updated methods from 2000 onwards.

- The AOAC methods were updated in 2005.

5. In every section the authors provided too much descriptive information on each study point, they should provide specific and concrete details on each parameter studied.

- The current version has been modified

6. Some data must provide as supplementary files.

- The suggested correction has been made.

---

## [Decision Letter · Decision Letter 2]

11 Aug 2021

A novel angiotensin I-converting enzyme inhibitory peptide derived from the trypsin hydrolysates of salmon bone proteins

PONE-D-21-03572R2

Dear Dr. Karnchanatat,

We’re pleased to inform you that your manuscript has been judged scientifically suitable for publication and will be formally accepted for publication once it meets all outstanding technical requirements.

Kind regards,

Shashi Kant Bhatia

Academic Editor

PLOS ONE

---

## [Editor Report · Acceptance letter]

13 Aug 2021

PONE-D-21-03572R2 

A novel angiotensin I-converting enzyme inhibitory peptide derived from the trypsin hydrolysates of salmon bone proteins 

Dear Dr. Karnchanatat:

I'm pleased to inform you that your manuscript has been deemed suitable for publication in PLOS ONE. Congratulations! Your manuscript is now with our production department. 

Kind regards, 

on behalf of

Dr. Shashi Kant Bhatia 

Academic Editor

PLOS ONE